# Characterization of binding kinetics and intracellular signaling of new psychoactive substances targeting cannabinoid receptor using transition-based reweighting method

Soumajit Dutta[1], Diwakar Shukla[1,2,3,4,5]*

[1]Department of Chemical and Biomolecular Engineering, University of Illinois at Urbana-Champaign, Urbana, United States; [2]Department of Chemistry, University of Illinois at Urbana-Champaign, Urbana, United States; [3]Center for Biophysics and Quantitative Biology, University of Illinois at Urbana-Champaign, Urbana, United States; [4]Department of Bioengineering, University of Illinois at Urbana-Champaign, Urbana, United States; [5]Cancer Center at Illinois, University of Illinois at Urbana-Champaign, Urbana, United States

*For correspondence:
diwakar@illinois.edu

**Competing interest:** The authors declare that no competing interests exist.

## eLife Assessment

A combination of molecular dynamics simulation and state-of-the-art statistical post-processing techniques provided **valuable** insight into GPCR-ligand dynamics. This manuscript provides **solid** evidence for differences in the binding/unbinding of classical cannabinoid drugs from new psychoactive substances. The results could aid in mitigating the public health threat these drugs pose.

**Abstract** New psychoactive substances (NPS) targeting human cannabinoid receptor 1 pose a significant threat to society as recreational abusive drugs that have pronounced physiological side effects. These greater adverse effects compared to classical cannabinoids have been linked to the higher downstream $\beta$-arrestin signaling. Thus, understanding the mechanism of differential signaling will reveal an important structure-activity relationship essential for identifying and potentially regulating NPS molecules. In this study, we simulate the slow (un)binding process of NPS MDMB-Fubinaca and classical cannabinoid HU-210 from $CB_1$ using multi-ensemble simulation to decipher the effects of ligand binding dynamics on downstream signaling. The transition-based reweighing method is used for the estimation of transition rates and underlying thermodynamics of (un)binding processes of ligands with nanomolar affinities. Our analyses reveal major interaction differences with transmembrane TM7 between NPS and classical cannabinoids. A variational autoencoder-based approach, neural relational inference (NRI), is applied to assess the allosteric effects on intracellular regions attributable to variations in binding pocket interactions. NRI analysis indicates a heightened level of allosteric control of NPxxY motif for NPS-bound receptors, which contributes to the higher probability of formation of a crucial triad interaction ($Y^{7.53}$-$Y^{5.58}$-$T^{3.46}$) necessary for stronger $\beta$-arrestin signaling. Hence, in this work, MD simulation, data-driven statistical methods, and deep learning point out the structural basis for the heightened physiological side effects associated with NPS, contributing to efforts aimed at mitigating their public health impact.

**Figure 1.** Classification of cannabinoid agonists. (**A**) Molecules derived from cannabis plants (phytocannabinoids) (**B**) endogenous agonists (Endocannabinoids) (**C**) synthetically designed molecules (Synthetic cannabinoids). Synthetic cannabinoids can be further classified based on scaffolds (phytocannabinoid analogues and endocannabinoid analogues or new psychoactive substances). Common pharmacophore groups of the ligands are shown in different colors. For phytocannabinoids and phytocannabinoid synthetic analogues, tricyclic benzopyran group and alkyl chains are colored in red and blue, respectively. Polar head group, propyl linker, polyene linker, and tail group of endocannabinoid and endocannabinoid analogues are colored with green, yellow, red, and orange, respectively. Linked, linker, core, and tail group of new psychoactive substances are colored with green, yellow, red, and orange, respectively.

The online version of this article includes the following figure supplement(s) for figure 1:

**Figure supplement 1.** Atom numbering scheme of classical cannabinoid.

**Figure supplement 2.** Pharmacophore components and representative new psychoactive substances (NPS) scaffolds.

## Introduction

Cannabinoid receptor 1 (CB$_1$), which is majorly expressed in the central nervous system (CNS) belongs to the class A G-protein coupled receptor (GPCR) family proteins (*Hua et al., 2016*; *Mackie, 2008*; *Zou and Kumar, 2018*; *Dutta and Shukla, 2023*). GPCRs are expressed in the cellular membrane and help transduce chemical signals from the extracellular to the intracellular direction with the help of the downstream signaling proteins (G-proteins and $\beta$-arrestin) (*Rosenbaum et al., 2009*; *Latorraca et al., 2017*; *Weis and Kobilka, 2018*). In addition, GPCRs are the largest family of drug targets due to their substantial involvement in human pathophysiology and druggability (*Hauser et al., 2017*; *Yang et al., 2021*). Significant research efforts have been invested in the discovery of drugs targeting CB$_1$, which helps to maintain homeostasis in neuron signaling and physiological processes (*Smith et al., 2010*; *An et al., 2020*).

Initial drug discovery efforts, especially the design of synthetic agonists, were based on modifying the scaffolds of phytocannabinoids (e.g. $\Delta^9$-Tetrahydrocannabinol, cannabinol) and endocannabinoids

(e.g. Anandamide, 2-arachidonoylglycerol) (*Figure 1*; *Pertwee, 2006*; *Pertwee and Ross, 2002*; *Pertwee et al., 2010*). The synthetic molecules, which maintain the aromatic, pyran, and cyclohexenyl ring of the most common psychoactive phytocannabinoid $\Delta^9$-THC, are known as classical cannabinoids (*Figure 1—figure supplement 1*; *Razdan, 2009 Madras, 2018*; *Dutta et al., 2022a*). However, the pharmacological potential of these molecules was diminished due to their psychological and physiological side effects ('tetrad' side effect) (*Moore and Weerts, 2022*; *Wang et al., 2020*; *Tummino et al., 2023*). One such example of a synthetic cannabinoid is 1,1-Dimethylheptyl-11-hydroxy-tetrahydrocannabinol (commonly known as HU-210), which is a Schedule I controlled substance in the United States (*Farinha-Ferreira et al., 2022*).

Apart from the canonical structures of synthetic cannabinoids, molecules with diverse scaffolds were also synthesized through structure-activity studies (*Wiley et al., 2016*; *Schoeder et al., 2018*; *Walsh and Andersen, 2020*). However, these molecules also lacked any pharmacological importance due to psychological side effects (*Akram et al., 2019*; *Worob and Wenthur, 2020*). Due to the diverse structures and psychological effects, these molecules became unregulated substitutes for traditional illicit substances (*Peacock et al., 2019*). These synthetic cannabinoids belong to a class of molecules known as NPS as these molecules are not scheduled under the Single Convention on Narcotic Drugs (1961) or the Convention on Psychotropic Substances (1971) (*Peacock et al., 2019*; *Madras, 2016*). Synthetic cannabinoids make up the largest category of NPS molecules (*Shafi et al., 2020*; *Alam and Keating, 2020*). NPS creates a significant challenge for drug enforcement agencies, as they appeal to drug users seeking 'legal highs' to avoid the legal consequences of using traditional drugs and to be undetectable in drug screenings (*Worob and Wenthur, 2020*).

The molecular structures of NPS synthetic cannabinoids consist of four pharmacophore components: linked, linker, core, and tail groups (*Worob and Wenthur, 2020*; *Potts et al., 2020*). The core usually consists of aromatic scaffolds (e.g. indole, indazole, carbazole, benzimidazole) (*Figure 1—figure supplement 2*; *Schoeder et al., 2018*). The tail and linker groups are connected to the core. In the tail group, long alkyl chain-like scaffolds are ubiquitous in most NPSs; however, molecules with bulkier cyclic chains (e.g. AB-CHMINACA) are also present (*Potts et al., 2020*). Frequently encountered scaffolds in linker groups are methanone, ethanone, carboxamide, and carboxylate ester groups (*Hill et al., 2018*). The linker acts as a bridge between the core and the linked group. In the initial NPS synthetic cannabinoids, the linked group included polyaromatic rings; however, non-cyclic linked groups have also been identified in NPS recently (*Schoeder et al., 2018*; *Potts et al., 2020*). Structural diversity in every component, while maintaining high binding affinity and potency for $CB_1$ make these molecules easier for drug manufacturers and harder to ban by drug enforcement agencies (*Banister et al., 2015a*; *Ametovski et al., 2020*; *Cannaert et al., 2020*; *Banister et al., 2015b*).

The use of NPS synthetic cannabinoids has been found to cause more physiological side effects than traditional cannabinergic 'tetrad' side effects (*Tai and Fantegrossi, 2014*). These side effects include tachycardia, drowsiness, dizziness, hypertension, seizures, convulsions, nausea, high blood pressure, and chest pain (*Tai and Fantegrossi, 2014*; *Finlay et al., 2019*). For instance, Gatch and Forster have shown that the high concentrations of AMB-FUBINACA, the molecule which caused 'zombie outbreak' in New York, induced tremors (*Gatch and Forster, 2019*; *Adams et al., 2017*). A recent biochemical study has linked these discriminatory effects with the differential signaling of β-arrestin (*Finlay et al., 2019*). According to Finlay et al., NPS shows higher β-arrestin signaling compared to the classical cannabinoids, which has also been confirmed by other β-arrestin signaling studies (*Finlay et al., 2019*; *Grafinger et al., 2021*). However, a mechanistic understanding of these differential downstream signaling effects between NPS and classical cannabinoids is still missing.

Mutagenesis studies have shown that the conserved NPxxY motif of $CB_1$ have a larger role in downstream β-arrestin signaling than G-protein signaling (*Leo et al., 2023*; *Liao et al., 2023*). Recently published MDMB-FUBINACA bound $CB_1$-β-arrestin-1 complex structure also points out the importance of the unique triad interaction ($Y397^{7.53}$-$Y294^{5.58}$-$T210^{3.46}$) involving NPxxY motif in β-arrestin-1 signaling (*Liao et al., 2023*). However, structural comparison of the classical cannabinoid (AM841) and NPS (MDMB-FUBINACA) bound active $CB_1$-$G_i$ complex shows a conformationally similar NPxxY motif (*Figure 2*; *Krishna Kumar et al., 2019*; *Hua et al., 2020*). In light of these experimental observations, it can be inferred that higher β-arrestin signaling stems from higher dynamic propensity of triad interaction formation for NPS-bound $CB_1$. We hypothesized that distinct orthosteric pocket interactions for

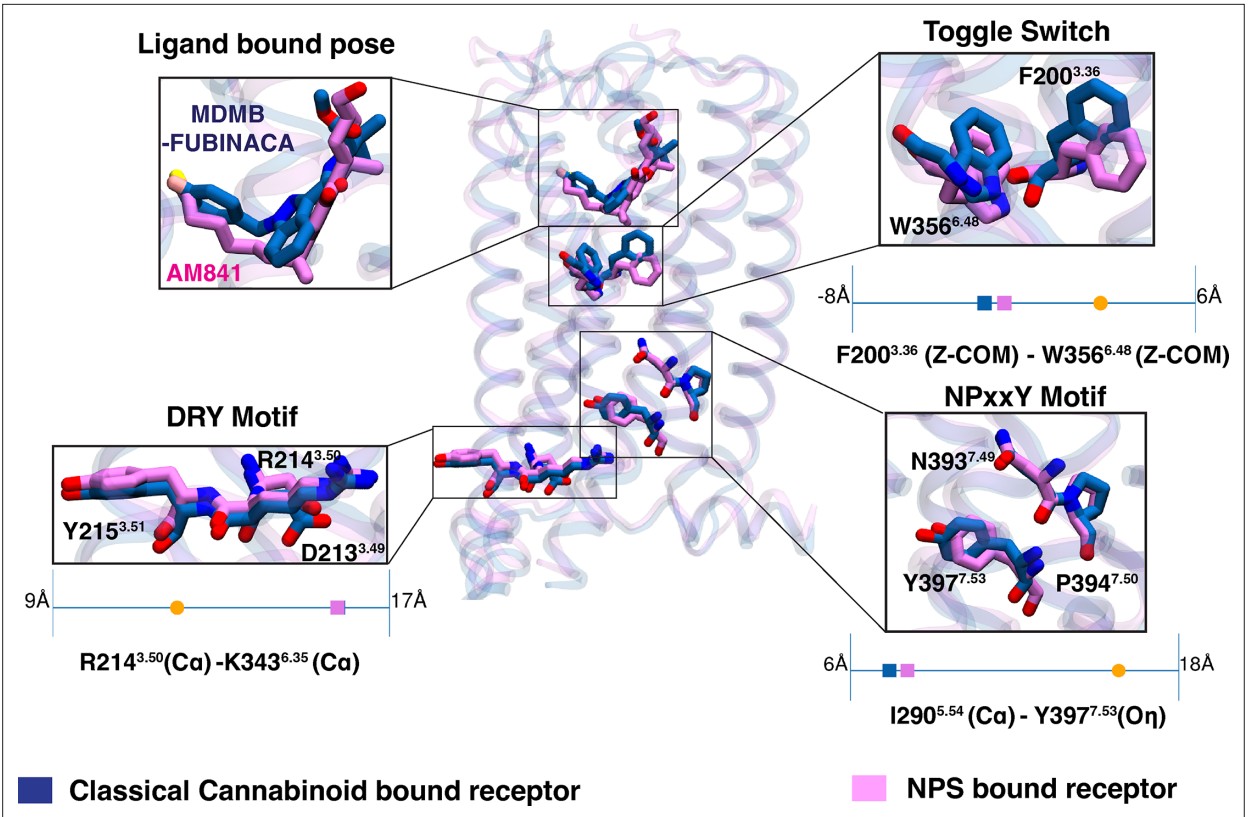

**Figure 2.** Structural comparison between new psychoactive substances (NPS) bound and classical cannabinoid bound $CB_1$. NPS bound $CB_1$ (PDB ID: 6N4B, *Krishna Kumar et al., 2019* color: Blue) structure is superposed with the classical cannabinoid bound $CB_1$ (PDB ID: 6 KPG, *Hua et al., 2020* color: Purple). Both structures are in $G_i$ bound active state. Proteins are shown in transparent cartoon representation. Structural comparison of conversed activation matrices (Toggle switch, DRY motif, and NPxxY motif) and ligand poses are shown as separate boxes. Quantitative values of the activation metrics for both active structures are compared as scatter points on 1-D line with the $CB_1$ inactive structure (PDB ID: 5TGZ, *Hua et al., 2016* color: orange). These quantitative measurements were discussed in *Dutta and Shukla, 2023*.

NPS and classical cannabinoids cause differential allosteric modulation of intracellular dynamics that facilitate triad interaction.

To study these distinct dynamic effects, we compared the (un)binding of the classical cannabinoid (HU-210) and NPS (MDMB-FUBINACA) from the receptor binding site. These molecules have nano-molar affinities. Obtaining the initial pathway of ligand unbinding from unbiased sampling will be computationally expensive. Therefore, a well-tempered metadynamics approach was used to sample the unbinding event, where a time-dependent biased potential is deposited for the faster sampling of the metastable minima along the pathway (*Barducci et al., 2008*). However, a detailed character-ization of the unbinding processes is only possible through the thermodynamics and kinetics esti-mation of intermediate states. Thus, a transition operator-based approach is needed, which helps to estimate the transition timescale between the states and the stationary density of each state. Estimation from these approaches usually depends on the equilibrium between the local states, which can only be maintained by reversible sampling. For high-affinity ligands like MDMB-FUBINACA and HU-210, reversible sampling is expensive as ligands move from high energy unbound states to lower energy bound states irreversibly. Hence, we implemented a transition operator approach named the transition-based reweighting analysis (TRAM) method, which can tackle this lack of local equilibrium between states by combining unbiased and biased approaches (*Wu et al., 2016*). TRAM has been used in in different simulation studies for estimating thermodynamics and kinetics of processes that have high free energy barriers. For example, TRAM have been utilized for characterization of small molecule and peptide (un)binding processes (*Wu et al., 2016*; *Paul et al., 2017*; *Ge et al., 2021*; *Spiriti et al., 2022*; *Ge and Voelz, 2022*), protein dimerization (*Meral et al., 2018*), ion transportation (*Hu et al., 2019*). To implement TRAM for our study, extensive sampling of the (un)binding process

of both ligands was performed using a combination of umbrella sampling and unbiased simulations from the pathway obtained using metadynamics (see Methods section) (*Kästner, 2011*). We showed that TRAM can produce consistent kinetic estimation with less unbiased simulation data compared to traditional methods like the Markov state model (*Prinz et al., 2011*).

Based on estimates of thermodynamics and kinetics, it was observed that both NPS and classical cannabinoids have similar unbinding pathways. However, their unbinding mechanisms differ due to the aromatic tail of the MDMB-FUBINACA compared to the alkyl side chain of HU-210. Furthermore, dynamic interaction calculations reveal a major difference with TM7 between NPS and classical cannabinoid. Specifically, the hydroxyl group in the benzopyran moiety of HU-210 forms much stronger polar interactions with $S383^{7.39}$ compared to the carbonyl oxygen of the linker group in MDMB-FUBINACA. MD simulations of other classical cannabinoids and NPS molecules bound to $CB_1$ also support these significant interaction differences. The ligand binding effect in intracellular signaling was estimated by measuring the probability of triad formation in the intracellular region. NPS-bound $CB_1$ shows higher probability of forming triad interaction compared to the classical cannabinoids, which supports the experimental observations of high $\beta$-arrestin signaling of NPS-bound receptors. To validate that the triad formation is indeed caused by the binding pocket interaction differences between the two ligands, allosteric strength binding pocket residues and NPxxY motif was estimated with the deep learning technique, Neural relational inference (NRI) (*Zhu et al., 2022a*). NRI network revealed that binding pocket residues of NPS-bound ensemble have higher allosteric weights for the NPxxY motif compared to classical cannabinoids. These analyses validate our hypothesis that the differential dynamic allosteric control of the NPxxY motif might lead to the $\beta$-arrestin signaling for different ligands. This study provides a foundation for additional computational and experimental research to enhance our understanding of the connection between ligand scaffolds and downstream signaling. This knowledge will assist drug enforcement agencies in proactively banning these molecules and inform policies that can protect individuals from the effects of abuse.

## Results and discussion

### Metadynamics simulations capture the unbinding paths of NPS and classical cannabinoids

The representative classical cannabinoid and NPS selected for this study are HU-210 and MDMB-FUBINACA (*Farinha-Ferreira et al., 2022*). Compared to $\Delta^9$-THC, HU-210 has an extra hydroxyl group in the C-11 position and a 1',1'-Dimethylheptyl group instead of a pentyl side chain (*Figure 1A and C*). MDMB-FUBINACA is a derivative of AB-FUBINACA, which was originally developed by Pfizer (*Figure 1C*; *Krishna Kumar et al., 2019*). These ligands binds to $CB_1$ receptor with nanomolar affinities (MDMB-FUBINACA $K_i$: 1.14 nM; *Gamage et al., 2018*; HU-210 $K_i$: 0.61 nM *Pertwee et al., 2010*; *Stern and Lambert, 2007*).

Metadynamics simulation is a biased sampling method and has been widely used in protein-ligand binding and unbinding studies, as preexisting knowledge of the pathway is not necessary for performing these simulations (*Ibrahim and Clark, 2019*; *Saleh et al., 2017*; *Mahinthichaichan et al., 2021*; *Saleh et al., 2018*). In metadynamics, a time-dependent biased potential is deposited into the sampling process for the ligand to get out of stable minima at a faster pace (*Barducci et al., 2011*). Here, two replicas of well-tempered metadynamics were performed to capture the unbinding pathways of MDMB-FUBINACA and HU-210 from the ligand-bound receptors, whose starting structures were obtained from a cryo-EM complex (PDB ID: 6N4B *Krishna Kumar et al., 2019*) and docking, respectively (*Figure 3—figure supplement 1*). The commonly used collective variables were selected for metadynamics simulations: (1) z component distance between the center of mass of ligand and residue in the ligand binding pocket ($W356^{6.48}$), and (2) Contact number with the ligand heavy atom and $\alpha$ carbon of all binding pocket residues (*Equation 4*).

The z-component distance was plotted against the RMSD of the ligands from the bound pose, which indicates that ligands follow a similar pathway for each replica (*Figure 3—figure supplement 2A and C*). It is observed that the dissociation happens via the opening formed by TM2, TM3, ECL2, and N-terminus for both ligands (*Figure 3*). We also performed unbinding simulations using well-tempered metadynamics parameters (bias height, bias deposition rate and bias factor) to confirm the existence of alternative pathways (*Figure 3—figure supplement 3*). However, the simulations show

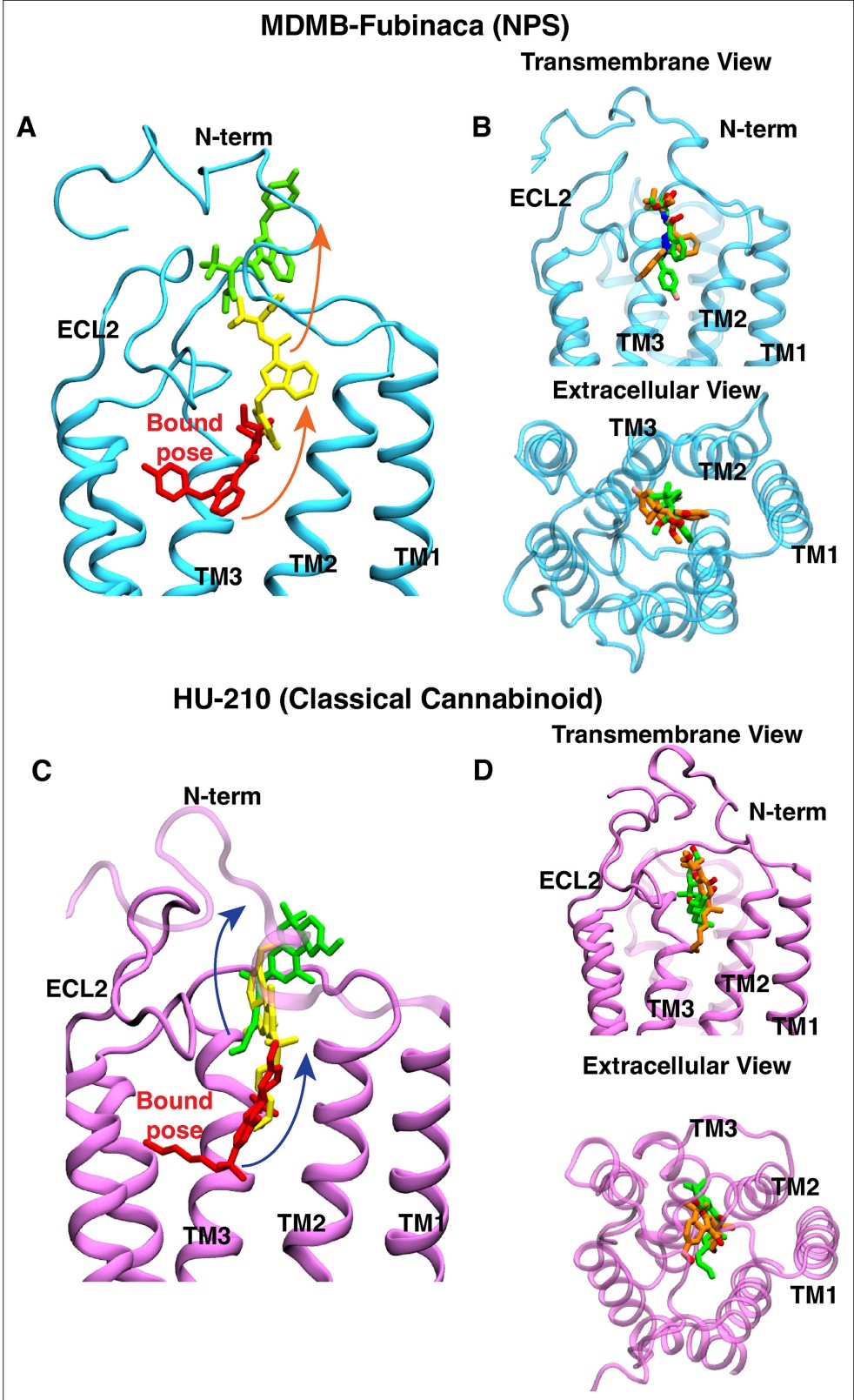

**Figure 3.** Ligand unbinding pathways for MDMB-FUBINACA and HU-210. The ligands MDMB-FUBINACA (**A**) and HU-210 (**C**) are depicted in three distinct stages along their unbinding pathways, as determined by well-tempered metadynamics simulations. The ligands are illustrated using stick representations, with each stage represented by a different color to indicate the progression from the bound (color: red) to the unbound state (color: green).

*Figure 3 continued on next page*

*Figure 3 continued*

Representative ligand positions from an intermediate state are shown in yellow. Additionally, the superposition of representative frames of an intermediate stage of the unbinding process is shown, where MDMB-FUBINACA (**B**) and HU-210 (**D**) are dissociating from the receptor. The frames are obtained from two different well-tempered metadynamics simulation replicas and are shown with different colors (green and orange). Both transmembrane (left panel) and extracellular (right panel) views are displayed. Proteins are represented as cartoons.

The online version of this article includes the following figure supplement(s) for figure 3:

**Figure supplement 1.** Comparison of cryo-EM and docked structures of two classical cannabinoids.

**Figure supplement 2.** Characterization of ligand binding pathways.

**Figure supplement 3.** The unbinding simulation with well-tempered metadynamics with different parameters.

---

that ligands follow the similar pathway for all metadynamics runs. These observations indicate that the pathway may be the minimum free energy pathway for the ligand unbinding in $CB_1$. Previous metadynamics binding simulation of another cannabinoid ligand also points to a similar pathway (*Saleh et al., 2018*). Reweighted probability density obtained from metadynamics calculation shows one highly dense region in the pocket, depicting the stability of the bound pose of the ligands (*Figure 3—figure supplement 2B and D*). However, time-dependent external force applied during the metadynamics makes the sampling in the orthogonal direction of the CVs less extensive. Thus, the biased simulation might not sample some protein-ligand interactions that helps to characterize intermediate states. To properly characterize intermediate transition states during the unbinding process, discrete kinetic models based on extensive unbiased simulations have been used. These unbiased simulations are often initiated from the pathways derived from the initial, limited sampling obtained through biased simulations (*Paul et al., 2017*; *Sun et al., 2018*; *Abella et al., 2020*) (discussed below).

## Comparison of thermodynamics and kinetics estimates from Markov state model and transition-based reweighting analysis method

MSM and TRAM are both postprocessing techniques for estimating the kinetics and thermodynamics of underlying physical processes observed in MD simulation. MSM is applied to reversible equilibrium simulations, whereas TRAM estimations can be obtained from multi-ensemble simulations (combination of biased and unbiased simulations). The MSM depends on the local equilibrium between the Markovian states, which is also known as detailed balance. However, reversible local sampling becomes challenging with short parallel trajectories when the free energy difference between two local Markovian states is high. In those cases, reversibility is still assumed by forcing the detailed balance when estimating the transition probability matrix (*Prinz et al., 2011*). This leads to the incorrect estimation of the unbinding kinetics due to limited sampling from the stable bound state to the high energy unbound states (*Wu et al., 2016*). Refining the state discretization (i.e. increasing the number of states) may resolve the issue. However, refined state discretization sometimes decreases the statistically significant transition count between all states, decreasing the model certainty. TRAM was shown to solve this problem by combining biased and unbiased simulations (see Methods section). Biased simulations (e.g. replica exchange, umbrella sampling) help to enhance the local sampling, either by increasing the temperature for faster sampling or by fixing collective variables with biased potential to have better sampling in orthogonal directions. It has been shown that compared to MSM, kinetics predicted using TRAM from the combination of biased and unbiased simulations are more aligned with the experiment results (*Wu et al., 2016*).

As unbinding of ligands with high binding affinity (nanomolar) are being studied here, asymmetric transitions might be observed along the pathway. Therefore, we compared the use of MSM and TRAM in estimating the kinetics and thermodynamics of the (un)binding process. For TRAM, unbiased simulation and umbrella sampling were run from the clusters in conformational ensemble obtained from metadynamics (*Figure 4—figure supplement 1*) (see Methods section for more details). For MSM estimation, only unbiased simulations starting from the metadynamics pathway were considered. MSM and TRAM featurization, building, and optimization process are discussed in detail in Methods section (*Figure 4—figure supplements 2–6*).

For thermodynamics comparison, standard free energy was estimated for the ligands considering volume correction (*Buch et al., 2011*). TRAM and MSM predictions of standard binding free energy

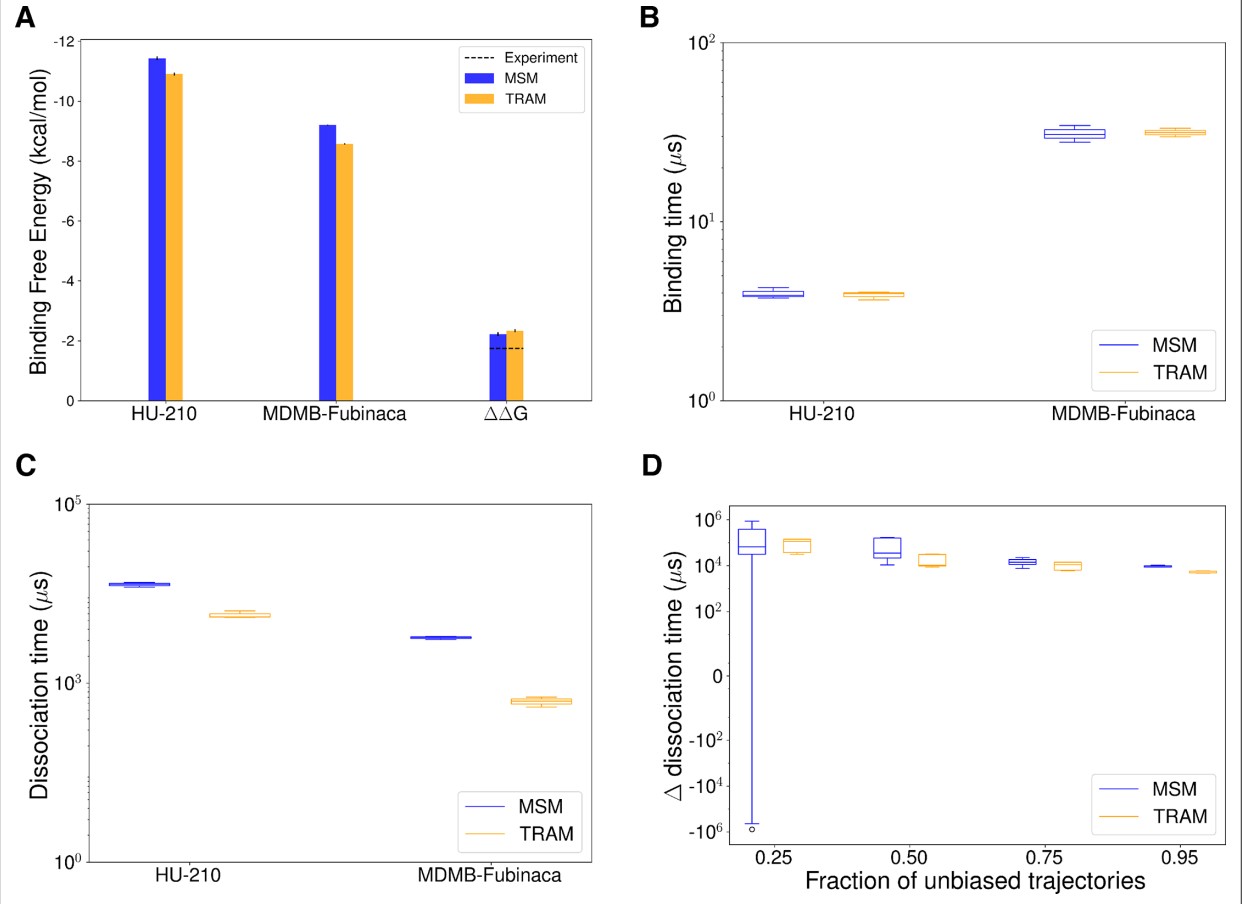

**Figure 4.** Comparison of thermodynamics and kinetics estimation of the unbinding process between MSM and transition-based reweighting analysis (TRAM). (**A**) The bar plot represents standard binding free energy for HU-210, MDMB-FUBINACA, and difference of standard binding free energy between the ligands. MSM and TRAM estimations are shown as blue and orange bars, respectively. Experimentally predicted values are shown as dotted line. (**B, C**) Binding (**B**) and dissociation (**C**) time for HU-210 and MDMB-FUBINACA are shown as box plots. (**D**) Difference in dissociation time of the two ligands is plotted as box plot against fraction of unbiased trajectories used for the estimation. These timescales were obtained from the mean free passage time calculation using transition path theory (TPT) with transition probabilities estimated from MSM (color: blue) and TRAM (color: orange). Errors were calculated using bootstrapping method with three bootstrapped samples.

The online version of this article includes the following figure supplement(s) for figure 4:

**Figure supplement 1.** Distance used to cluster the metadynamics sampled (un)binding pathway.

**Figure supplement 2.** Binding pocket residues considered in MSM and TRAM analysis.

**Figure supplement 3.** Implied timescale convergence with MSM lag time.

**Figure supplement 4.** Optimization of VAMP-2 scores.

**Figure supplement 5.** Chapman–Kolmogorov (C–K) test to judge Markovianity of MSM.

**Figure supplement 6.** Sampled raw probability density vs estimated weighted probability from MSM and transition-based reweighting analysis (TRAM).

**Figure supplement 7.** Convergence of $\Delta\Delta G$ calculation.

are within 0.6 kcal/mol of each other for each ligand (**Figure 4A**). Although the absolute binding free energy differs from the experimentally predicted value by approximately 3 kcal/mol, the relative estimated free energy ($\Delta\Delta G$) values are also within 0.6 kcal/mol of the experimentally determined values (**Figure 4—figure supplement 7**). Therefore, it indicates that with sufficient sampling, both MSM and TRAM converge to the same predictions of relative free energy.

We also compared the kinetics obtained from the MSM and TRAM. Kinetic measurements were performed with transition path theory (TPT), which uses transition probability matrix from MSM or TRAM to estimate mean free passage time between different states (see Methods section). Estimated binding times using TRAM and MSM match perfectly for both ligands (**Figure 4B**). The estimated

dissociation times are within one order of magnitude with each other (*Figure 4C*). These observations agree with the previously reported computational research, where experimentally comparable estimation of $k_{off}$ rates were shown to be more challenging compared to $k_{on}$ (*Wang et al., 2023*).

Further analyses were performed to compare these methods in the low-unbiased data regime. The difference between the dissociation time of ligands was measured with different amounts of unbiased data. It is observed that even with only 25% of original, unbiased data, TRAM can predict the kinetics within an order magnitude of the kinetics estimated with full dataset (*Figure 4D*). On the other hand, error in MSM predicted kinetics more rapidly compared to TRAM with lesser amount of unbiased data. A similar trend can be observed for ΔΔG prediction (*Figure 4—figure supplement 7*). Therefore, TRAM provides better predictions of thermodynamics and kinetics when less amount of unbiased data.

## Unbinding mechanism for new psychoactive substance

Although the binding position of the ligand and the overall binding pathway are similar for both the ligands, extensive biased and unbiased simulation analyzed by TRAM shows a significant difference in the unbinding mechanism of the ligands. To capture the unbinding pathway for MDMB-FUBINACA, we projected the TRAM weighted free energy landscape of the distance between the linked part of the ligand (Leucinate group) and TM5 with respect to the distance between the ligand tail group and TM7 (*Figure 5A*). The free energy landscape was divided into the non-overlapping intermediate macrostates to obtain better description of the unbinding process. In each macrostate, the contact frequency of ligand with binding pocket residues were calculated along with corresponding contact energies. A metastable minimum is observed for macrostate representing the bound pose of the ligand depicting the stability of the ligand (*Figure 5A*). In the bound pose, the major interactions form between the aromatic (F170$^{2.57}$, W279$^{5.43}$, F268$^{ECL2}$) and hydrophobic (L193$^{3.29}$) residues of the binding pocket (*Figure 6A and B*, *Figure 6—figure supplement 1B*).

The free energy landscape shows two probable mechanisms for MDMB-FUBINACA unbinding from the bound pose. The two pathways are differentiated by whether the linked or tail part of MDMB-FUBINACA dissociates first. One of the pathways, aromatic tail part of MDMB-FUBINACA moves away from TM5 and forms interactions with aromatic residues in TM2 (F170$^{2.57}$ and F174$^{2.61}$) (*Figures 5A and 6A, B*, *Figure 6—figure supplement 1*). This leads to the formation of intermediate metastable states, which we characterize as macrostate Intermediate state 1 (I1). This metastable minimum observed from I1 macrostate might be unique to the FUBINACA family of NPS synthetic cannabinoids as this family has the aromatic ring in tail group, unlike the long alkyl chain in other common synthetic cannabinoids. Along with the aromatic residues of TM2, major interaction with F268$^{ECL2}$ is maintained in macrostate I1 (*Figure 6A and B*). KL divergence analysis of the inverse distance distributions between two macrostates was conducted to highlight significant conformational changes. The bound and I1 macrostates show that only minor changes in the binding pocket residues, especially in TM2 are needed to accommodate MDMB-FUBINACA in this conformational state (*Figure 6C*). The interconversion timescale (MFPT) between the macrostates were obtained from the transition path theory. MFPT calculations show that both the timescales are similar with slightly higher timescales for the bound pose compared to I1 transition (20.6±2.3 μs) (*Figure 6B*). In this pathway, the ligand moves from I1 metastable state to space between N-terminus, TM2, TM3, and ECL2 before dissociating from the receptor (*Figure 6B*). This region between the unbinding ensemble has been characterized as macrostate I3 (*Figure 5A*). Contact analysis shows significant drop in ligand residue contacts with only aromatic residues in TM2, TM3, and ECL2 forming dominant interactions (*Figure 6A*, *Figure 6—figure supplement 1*). We further performed Kullback-Leibler divergence (K-L divergence) analysis between inverse distance of residue pairs of two macrostates to highlight the protein region that undergoes high conformational change with ligand movement (detailed discussion in Methods section). K-L divergence shows that ligand positioning in these particular regions causes relatively higher divergence on TM2 compared to I1 (*Figure 6C*). Kinetically, the transition from I1 to I3 (33.7±3.1 μs) is much slower compared to reverse transition (0.8±0.0 μs), validating the higher stability of the I1 compared to I3 macrostate (*Figure 6B*). According to the TPT analysis, breaking the aromatic interactions for complete dissociation of MDMB-FUBINACA requires ~371.9±40.2 μs, making it the slowest step in this pathway (*Figure 6B*).

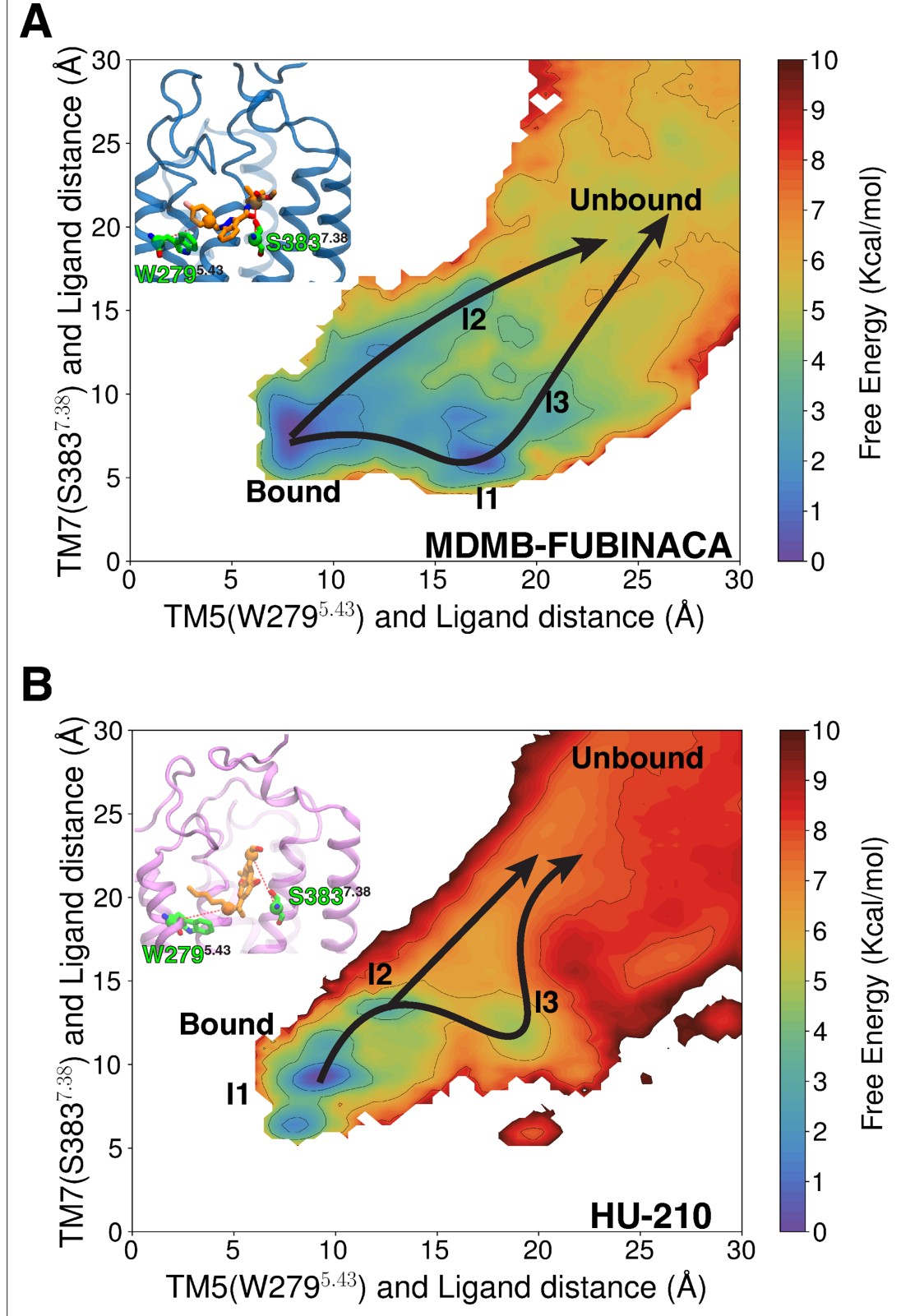

**Figure 5.** Transition-based reweighting analysis (TRAM)-weighted two-dimensional projection of unbinding free energy landscape for MDMB-FUBINACA (**A**) and HU-210 (**B**). For MDMB-FUBINACA, distance between TM5 (W279$^{5.43}$-Cα) and tail part of the ligand is plotted against the distance between TM7 (S383$^{7.39}$-Cα) and ligand-linked part. For HU-210, distance between the TM5 (W279$^{5.43}$-Cα) and tail is plotted against the TM7 (S383$^{7.39}$-Cα) and cyclohexenyl ring of the ligand. Measured distances are shown as red dotted lines in the inset figures. Macrostate positions are shown on the landscapes. Different mechanisms of (un)binding are shown with arrow on top of the free energy landscape.

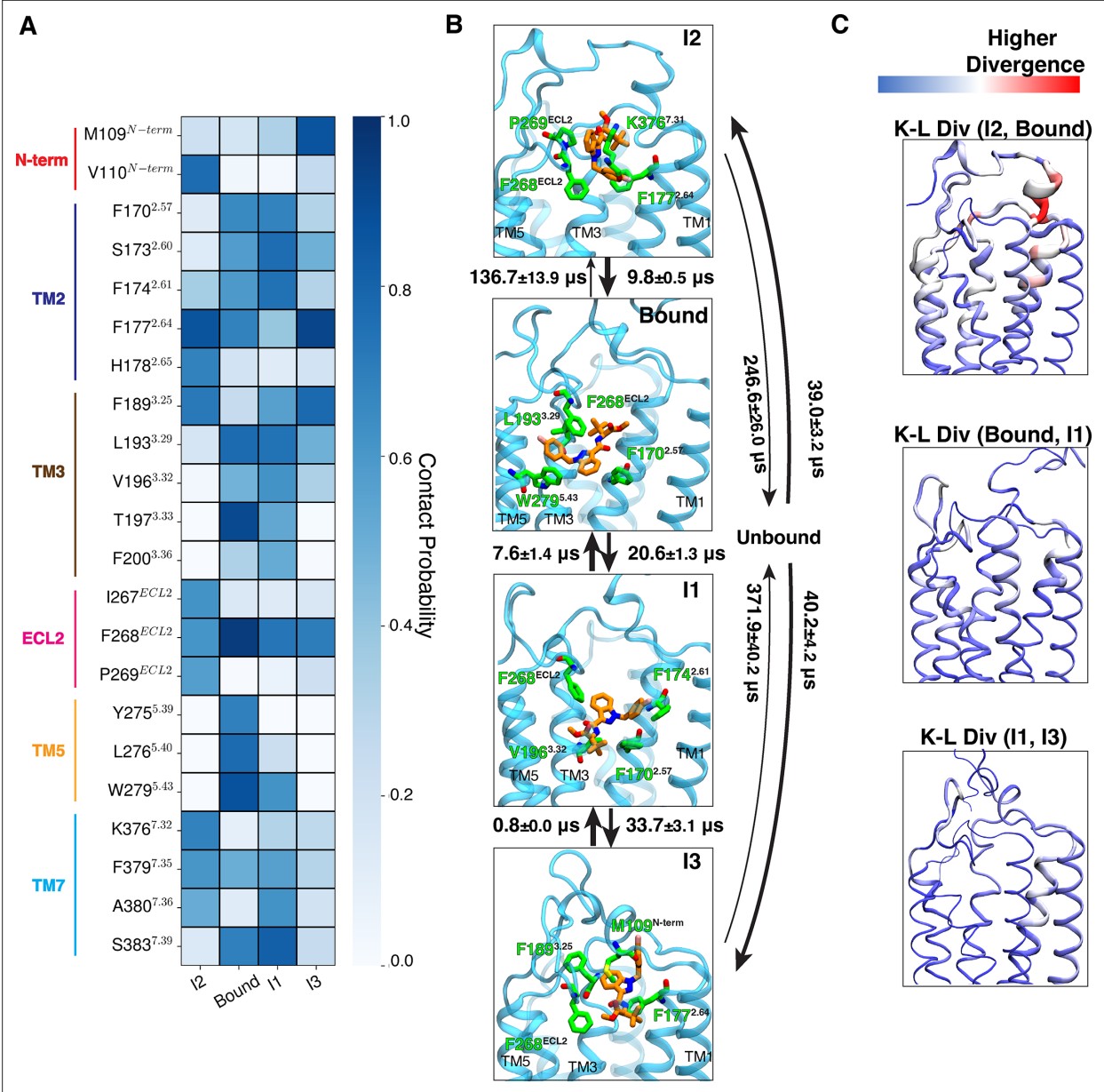

**Figure 6.** Mechanism of new psychoactive substances (NPS) MDMB-FUBINACA unbinding from $CB_1$. (**A**) The contact probabilities with binding pocket residues of MDMB-FUBINACA are shown as a heatmap for different macrostates, where ligand maintains contact with the receptor. Residues in different structural elements (loops and helices) are distinguished by distinct color bars. (**B**) Representative structures are shown where ligand (color: orange) and four residues (color: green) with highest interaction energies are shown as sticks. Proteins are shown as purple cartoon. Timescales between interstate transitions are shown as arrows. Arrow thickness is inversely proportional to the order of magnitude of the timescale. (**C**) Per residue K-L divergences between different states are shown with color (blue to red) and thickness (lower to higher) gradient. K-L divergences calculated on the inverse distance feature distributions were converted by residue basis by summing all the pair contributions corresponding to the residue. Thickness gradients are shown as rolling average to highlight a region of high K-L divergence. Errors in MFPT calculations were estimated based on three bootstrapped transition-based reweighting analysis (TRAM) calculations with randomly selected 95% of unbiased trajectories.

The online version of this article includes the following figure supplement(s) for figure 6:

**Figure supplement 1.** Contact probability and interaction energy calculations between MDMB-FUBINACA and $CB_1$ binding pocket residues for each macrostate.

**Figure supplement 2.** Root mean square deviation of the MDMB-FUBINACA in different macrostates.

In the other possible unbinding pathway, orientation of MDMB-FUBINACA in the pocket does not change compared to the bound pose. The linked part of the ligand moves to space between N-terminus, TM2, TM3, and ECL2 (*Figure 6B*). We label this macrostate as I2. In this state, we observe stable polar interaction with K376$^{7.32}$ and hydrophobic interactions with aromatic and other hydrophobic residues (F177$^{2.64}$, F268$^{ECL2}$, P269$^{ECL2}$) (*Figure 6A* and *Figure 6—figure supplement 1A*). However, free energy of this macrostate is higher than the bound pose, depicting higher entropic cost associated with this state. This can be shown by the higher intrastate RMSD of I3 compared to the bound pose (*Figure 6—figure supplement 2*). Transition timescale from the bound pose to the I2 (136.7±13.9 μs) is one order of magnitude higher compared to the reverse transition (9.8±13.9 μs) (*Figure 6B*). K-L divergence analysis also shows higher divergence in the extracellular region of TM2 and N-terminus compared to bound pose (*Figure 6C*). Dissociation of MDMB-FUBINACA from I2 to the bulk is faster compared to dissociation from I3 (246.6±26.0 μs) (*Figure 6B*). However, the overall kinetic barrier for dissociation from the binding pose for both unbinding mechanisms are relatively similar.

## Unbinding mechanism for classical cannabinoid

For capturing the classical cannabinoid (un)binding mechanism, distances from the two terminal scaffolds (cyclohexenyl and alkyl chain) to TM5 and TM7 were measured similar to the NPS (*Figure 5B*). The free energy landscape of the unbinding of the HU-210 shows the differences in the mechanism from MDMB-FUBINACA. Similar to MDMB-FUBINACA, the HU-210 unbinding landscapes were also divided into non-overlapping macrostates. Macrostate representing HU-210 bound pose shows a metastable energy minimum. Comparing the bound macrostate interactions of MDMB-FUBINACA, classical cannabinoid HU-210 shows higher interactions with TM7 residues (S383$^{7.39}$, F379$^{7.35}$) (*Figures 6A and 7A*, *Figure 6—figure supplement 1B* and *Figure 7—figure supplement 1B*). Previous experimentally determined structures of classical cannabinoid bound CB$_1$ have pointed out these conserved polar interactions of the hydroxyl group at C-1 position with S383$^{7.39}$ (*Hua et al., 2020*; *Hua et al., 2017*). Although MDMB-FUBINACA also maintain this polar interaction with carboxylic oxygen, the interaction energy for the HU-210 is much higher, depicting the importance of this conserved residue in stabilizing classical cannabinoids (*Figure 6B*). Mutagenesis studies also support this difference in interaction with S383$^{7.39}$ between the classical cannabinoids with hydroxyl group (HU-210) and CB$_1$ ligands, which have carboxylic oxygen in the equivalent position (WIN-55,212–2) (*Kapur et al., 2007*; *Sitkoff et al., 2011*). Alanine mutation of S383$^{7.39}$ have shown to decrease the ligand affinity and downstream efficacy of classical cannabinoids by orders of magnitude, while having minimal effect on WIN-55,212–2, which have carboxylic oxygen in the linked part as MDMB-FUBINACA (*Kapur et al., 2007*; *Sitkoff et al., 2011*). Other major interactions (F170$^{2.57}$ and F268$^{ECL2}$) in the bound pose are common between the two ligands (*Figures 6B and 7B*).

A relatively weaker metastable state is observed when the ligand moves relatively deeper (closer to TM5) inside the binding pocket. The flexible alkyl chain of HU-210 allows the ligand to have this deeper position (*Figure 7A and B*). Protein-ligand interaction analysis in the macrostate representing this region (I1) shows that hydroxyl group at C-11 forms a major polar interaction with H178$^{2.65}$ (*Figure 7—figure supplement 1A*). The bound and I1 macrostates are kinetically close, as indicated by the rapid interconversion between these states (*Figure 7B*). K-L divergence between the two states shows the highest divergence in extracellular TM2 and TM7, where major interaction switches have happened (*Figure 7C*).

Contrasting to MDMB-FUBINACA, only one pathway was discovered with classical cannabinoid cyclic scaffold departing from the receptor first. Major interactions that break when the ligand moves out of the binding pose to macrostate I2 is the polar interaction with S383$^{7.39}$ and hydrophobic interaction with aromatic F170$^{2.57}$ (*Figure 7A and B*, *Figure 7—figure supplement 1C*). Breaking of these bonds leads to larger kinetic barrier of approximately 39.5±1.4 μs (*Figure 7B*). In this macrostate, the HU-210 forms major interactions with aromatic residues F268$^{ECL2}$ and F177$^{2.64}$ and polar interactions with S173$^{2.60}$ and D176$^{2.63}$ (*Figure 7B*). From this pose, HU-210 either dissociates from the receptor or obtain another relatively weak stabilized state (I3) in the receptor. In I3, the alkyl chain of the ligand is flipped in the pocket and stabilized by aromatic residues in TM2, TM3, and ECL2 (*Figure 7B*). This transition from I2 to I3 (5.28±32.8 μs) kinetically much slower compared to the reverse transition (6.9±0.0 μs) (*Figure 7B*). From both I2 and I3 macrostates, the ligand can dissociate from the pocket and mean free passage time for these transitions appear to be in the millisecond timescale, which

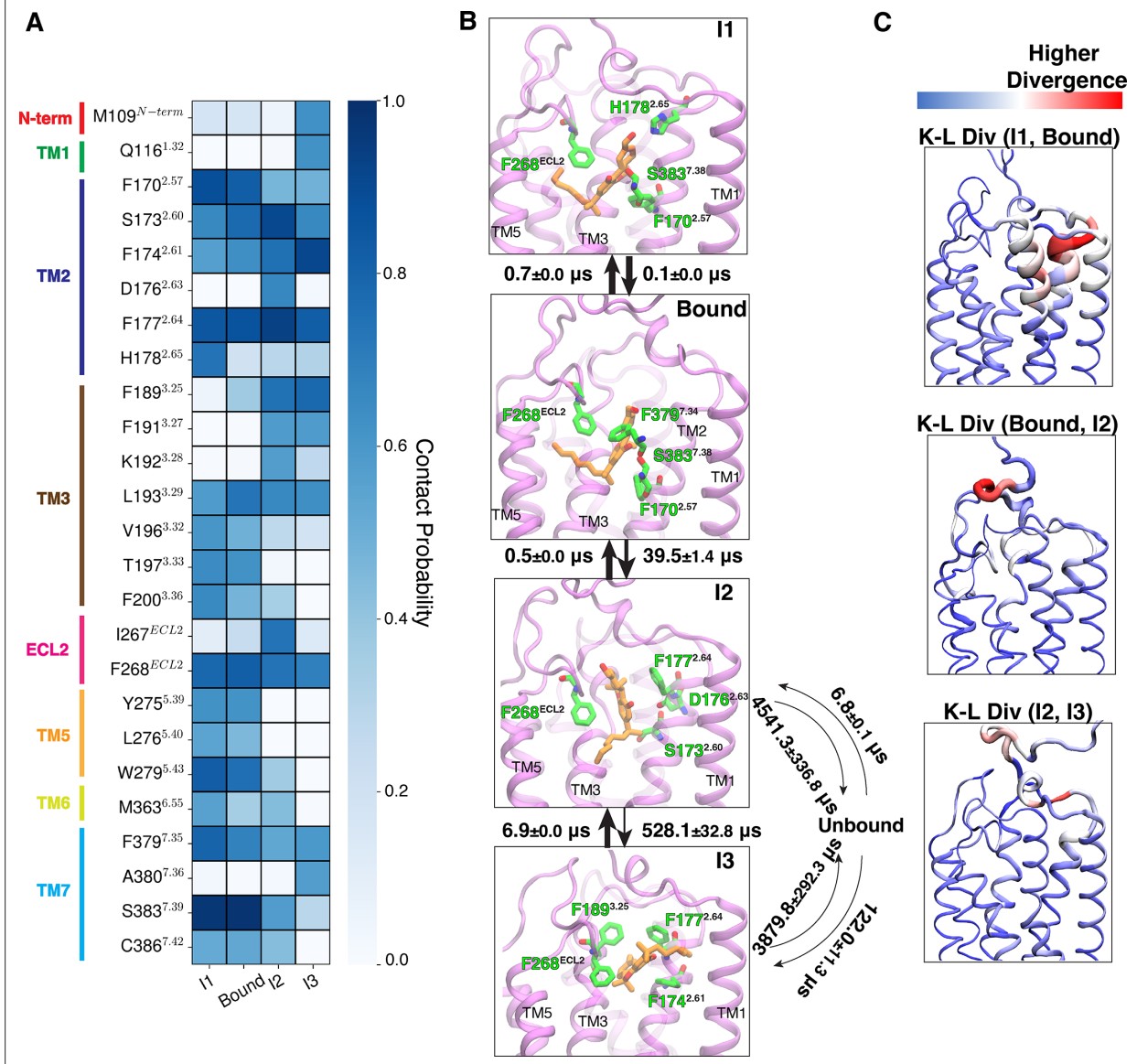

**Figure 7.** Mechanism of classical cannabinoid HU-210 unbinding from $CB_1$. (**A**) The contact probabilities with binding pocket residues of HU-210 are shown as a heatmap for different macrostates, where ligand maintains contact with the receptor. Residues in different structural elements (loops and helices) are distinguished by distinct color bars. (**B**) Representative structures are shown where ligand (color: orange) and four residues (color: green) with highest interaction energies are shown as sticks. Proteins are shown as purple cartoons. Timescales between interstate transitions are shown as arrows. Arrow thickness is inversely proportional to the order of magnitude of the timescale. Errors in MFPT calculations were estimated based on three bootstrapped transition-based reweighting analysis (TRAM) calculations with randomly selected 95% of unbiased trajectories. (**C**) Per residue K-L divergences between different states are shown with color (blue to red) and thickness (lower to higher) gradient. K-L divergences calculated on the inverse distance feature distributions were converted by residue basis by summing all the pair contributions corresponding to the residue. Thickness gradient are shown as rolling average to highlight a region of high K-L divergence.

The online version of this article includes the following figure supplement(s) for figure 7:

**Figure supplement 1.** Contact probability and interaction energy calculations between HU-210 and $CB_1$ binding pocket residues for each macrostate.

is one order of magnitude higher compared to the MDMB-FUBINACA unbinding (*Figure 6B*). This phenomenon supports the relatively high affinity of the classic cannabinoid HU-210 compared to the NPS MDMB-FUBINACA.

## Allosterically controlled distinct downstream signaling between new psychoactive substances and classical cannabinoids

As discussed in the previous section, major interaction differences between NPS MDMB-FUBINACA and classical cannabinoid HU-210 are observed in TM7. To support the universality of this observation, we performed unbiased MD simulation (1 μs each) of other NPS (AMB-FUBINACA, 5F-AMP, CUMYL-FUBINACA) and classical cannabinoids (AMG-41, JWH-133, O-1317) bound CB$_1$ (*Figure 8—figure supplement 1A*). Average distance of carbonyl oxygen of NPS molecules' linker group from S383$^{7.39}$ is compared to equivalent distance of hydroxyl group of classical cannabinoids' benzopyran ring. Larger mean distance in case of all NPS-bound CB$_1$ supports the universality of the weaker interaction between TM7 and NPS molecules (*Figure 8—figure supplement 1B*). This variation in binding pocket interactions might lead to differential allosteric control of the intracellular dynamics that facilitate triad interaction (Y397$^{7.35}$-Y294$^{5.58}$-T210$^{3.46}$) important for β-arrestin binding.

We adopted a data-driven deep learning network known as Neural relational inference (NRI) to validate our hypothesis of allosteric control. NRI network has an architecture of variational autoencoder. The encoder part of the network predicts the interactions between the residues from the trajectory dynamics, and the decoder predicts the trajectories from the interaction. With this network, we try to produce alpha carbon coordinates at $t + \tau$ from the coordinates at time t. In the process of regenerating the future coordinates, the latent space of the network learns the dynamic interactions between different residues in the protein. These interactions are calculated from the estimated posterior probability $q(z_{ij}|x)$. In this work, we trained the network with the NPS (MDMB-FUBINACA), and classical cannabinoid (HU-210) bound unbiased trajectories (Method Section) (*Figure 8—figure supplement 2*). Here, we compared the allosteric interaction weights between the binding pocket and the NPxxY motif which involves in triad interaction formation. Results show that each binding pocket residue in

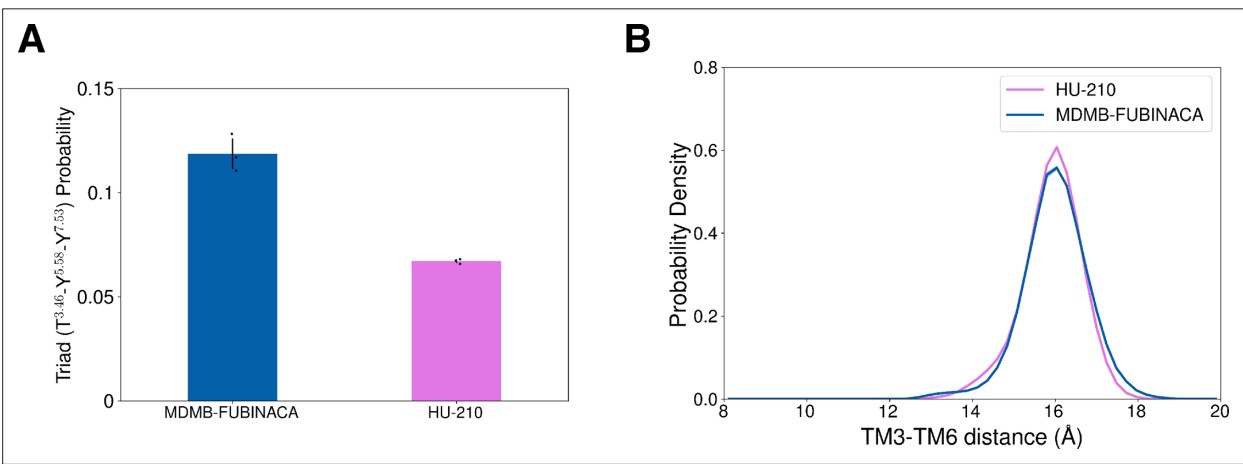

**Figure 8.** Dynamic conformational change in intracellular region of the CB$_1$ during ligand (un)binding. (**A**) Transition-based reweighting analysis (TRAM) weighted probabilities of triad interaction (Y397$^{7.53}$-Y294$^{5.58}$-T210$^{3.46}$) formation are plotted for HU-210 (color: purple) and MDMB-FUBINACA (color: blue) unbinding ensemble. If side-chain oxygen atoms of all three residues are within 5 Å of each other, triad interaction is considered to be formed. (**B**) TRAM weighted probability densities of TM3 (R214$^{3.50}$) and TM6 (K343$^{6.35}$) distance distribution are plotted for HU-210 (color: purple) and MDMB-FUBINACA (color: blue) unbinding ensemble. Error in the probability densities is estimated using a bootstrapping approach, where TRAM was built for three bootstrapped samples with 95% of total data.

The online version of this article includes the following figure supplement(s) for figure 8:

**Figure supplement 1.** Ligand interaction of other classical cannabinoids and new psychoactive substances (NPS) molecules with TM7.

**Figure supplement 2.** Training and validation losses during neural relational inference (NRI) network training.

**Figure supplement 3.** Neural relational inference (NRI)-based allosteric weight estimation.

**Figure supplement 4.** Mutual information-based allosteric weight estimation.

**Figure supplement 5.** Probability densities of pairwise distances of residues involved in triad interaction.

MDMB-FUBINACA bound ensemble shows higher allosteric weights with the NPxxY motif, indicating larger dynamic interactions between the NPxxY motif and binding pocket residues (*Figure 8—figure supplement 3*). To further validate our observations, we estimated allosteric weights between the binding pocket and the NPxxY motif by calculating mutual information between residue movements. Mutual information analysis reaffirms that allosteric weights between these residues are indeed higher for the MDMB-FUBINACA bound ensemble (*Figure 8—figure supplement 4*).

The probability of triad formation was estimated to observe the effect of the difference in allosteric control. TRAM weighted probability calculation showed that MDMB-FUBINACA bound $CB_1$ has the higher probability of triad formation (*Figure 8A*). Comparison of the pairwise interaction of the triad residues shows that interaction between $Y397^{7.53}$-$T210^{3.46}$ is relatively more stable in case of MDMB-FUBINACA bound $CB_1$, while other two interactions have similar behavior for both systems (*Figure 8—figure supplement 5A, B, and C*). Therefore, higher interaction between $Y397^{7.53}$ and $T210^{3.46}$ in MDMB-FUBINACA bound receptor causes the triad interaction to be more probable.

Furthermore, we also compared TM6 movement for both ligand-bound ensemble which is another activation metric involved in both G-protein and $\beta$-arrestin binding. Comparison of TM6 distance from the DRY motif of TM3 shows similar distribution for HU-210 and MDMB-FUBINACA (*Figure 8B*). These observations support that NPS binding causes higher $\beta$-arrestin signaling by allosterically controlling triad interaction formation.

## Conclusions

Synthetic cannabinoids were designed as a potential therapeutics to target cannabinoid receptors. However, major side effects of these ligands diminish their therapeutic potential. Although both classical cannabinoids and NPS synthetic cannabinoids have been abused as recreational drugs, later poses larger threats for the society due the chemical diversity of the NPS structures makes them harder to control from being abused. Furthermore, physiological studies have shown NPS targeting cannabinoid receptors lead to the dangerous physiological effects compared to 'tetrad' side effects associated with classical cannabinoids. Previous studies have related these side effects with the higher downstream $\beta$-arrestin signaling of NPS. Mutagenesis studies have shown NPxxY motif have larger role to play in $\beta$-arrestin signaling. In this work, we proposed that NPS and classical cannabinoid have distinct allosteric control on NPxxY motif when bound to orthosteric pocket of $CB_1$. In this hypothesis-driven study, we compared ligand-protein interactions of NPS MDMB-FUBINACA and classical cannabinoid HU-210 for $CB_1$ by studying their unbinding mechanism, and downstream signaling.

As both ligands are stable binders with nanomolar affinity, well-tempered metadynamics simulations were performed to obtain the initial unbinding pathway. These simulations were able to find similar pathways via the opening formed by TM2, TM3, N-terminus, and ECL2 which matches with previous metadynamics binding simulations for cannabinoid receptors. For the proper characterization of intermediate states, the unbinding processes were further extensively sampled using unbiased simulation and umbrella sampling.

Effectiveness of the post-processing techniques TRAM and MSM were compared in predicting the ligands with binding affinity and kinetics. MSM predicts the kinetics and thermodynamics from the eigendecomposition of the transition probability matrix. MSM assumes that the local equilibrium is maintained between the Markovian states. However, with limited sampling, this criterion may not valid between local high and low energy states. TRAM tries to solve this issue by combining biased and unbiased simulation, where biased simulations enhances the local sampling to maintain the equilibrium. We observed that with sufficient data, both methods performed in a similar way in estimating the standard binding free energy. The relative free energy estimated by both methods matches the experimental result within 0.6 kcal/mol. With a lesser amount of unbiased data, TRAM predictions of kinetics and thermodynamics remain more consistent than the MSM as the biased simulations help to maintain local equilibrium.

TRAM estimated thermodynamics helped to decipher the differences between the unbinding of NPS MDMB-FUBINACA and classical cannabinoid HU-210. First, for MDMB-FUBINACA, a larger conformational change is observed within the pocket. A metastable intermediate state is observed when the aromatic tail of FUBINACA flip inside the pocket and reorient itself close to the aromatic residues of TM2. It was observed that both linked part and tail part of the ligands can lead the dissociation of the ligand from the receptor. Second, for HU-210, conserved cyclic group leads to the

dissociation from the receptor. It supports previous simulation where the alkyl side chain of the ligand binds to the receptor first (*Dutta et al., 2022a*). Third, aromatic residues in the pocket (F2688[ECL2], F170[2.57]) form major interactions with both HU-210 and MDMB-FUBINACA. Major differences in protein-ligand interactions were observed in TM7. Stronger interactions were observed for the classical cannabinoid HU-210 with TM7, especially polar interaction with S383[7.39] and hydrophobic interaction with F379[7.35] compared to MDMB-FUBINACA. This interaction pattern was consistent across other NPS and classical cannabinoids, indicating a universal difference in how these two groups of compounds interact with TM7.

Finally, we demonstrated that the variation in binding pocket interaction leads to the distinct dynamic allosteric communications in the intracellular region. Allosteric communication strength was measured by the variational autoencoder (NRI). NRI network learns the dynamic interactions between residues in the latent space by learning to reconstruct the dynamics. Dynamic allostery measured by the posterior probability of VAE shows that higher allosteric weights from the binding pocket residues to the NPxxY motif region for MDMB-FUBINACA bound $CB_1$ increases the probability of triad interaction formation. Since the triad interaction is crucial for $\beta$-arrestin signaling, these findings align with experimental observations of enhanced $\beta$-arrestin signaling in NPS-bound receptors. Overall, this data-driven computational study helps us to distinguish between the receptor-protein interaction, unbinding mechanism and downstream signaling NPS compared to other classical cannabinoids.

## Methods
### System preparation

For NPS unbinding simulation, G-protein bound active structure (PDB: 6N4B *Krishna Kumar et al., 2019*) was selected as the initial structure. G-protein subunits and non-protein residues other than orthosteric ligand MDMB-Fubinaca were removed from the PDB structure file. Missing residues in ICL3 (21 residues, 314–334) and ECL2 (6 residues, 258–263) were modeled sequentially using Remodel protocol of Rosetta loop modeling (*Stein and Kortemme, 2013*; *Alford et al., 2017*). In each step, the remodeled structure with least energy was further refined using kinematic closure protocol (*Mandell et al., 2009*).

Starting 108 residues from $CB_1$ N-terminus were also missing from the cryo-EM structure. However, it is not feasible to model the entire N-terminus because of the following two reasons (*Stein and Kortemme, 2013*). First, a proper template is not available for modeling N-terminus regions as most of the class A GPCRs do not contain large N-terminus (*Pándy-Szekeres et al., 2018*). Second, it is challenging to model these large numbers of residues accurately with template-free *ab initio* modeling because of the combinatorial expansions of conformational space. Therefore, the closest 20 residues were modeled as membrane proximal regions of the N-terminus were shown to be important for $CB_1$ signaling by allosterically modulating ligand affinity (*Fay and Farrens, 2013*). Furthermore, $\Delta 89CB_1$ ($CB_1$ with first 88 residues truncated in N-terminus) have similar ligand binding affinity compared to $CB_1$ with full sequence (*Andersson et al., 2003*). Modeling of this membrane proximal region was also performed using the Remodel protocol of Rosetta loop modeling. A distance constraint is added during this modeling step between C98[N-term] and C107[N-term] to create the disulfide bond between the residues (*Fay and Farrens, 2013*; *Richter et al., 2011*).

As the cryo-EM structure of bound MDMB-FUBINACA was known, ligand coordinate of MDMB-FUBINACA was added to the modeled PDB structure. The 'Ligand Reader & Modeler' module of CHARMM-GUI was used for ligand (e.g. MDMB-Fubinaca) parameterization using CHARMM General Force Field (CGenFF; *Vanommeslaeghe et al., 2010*). The ligand-bound receptor was embedded in the bilayer membrane and salt solution (extracellular and intracellular region) using CHARMM-GUI (*Jo et al., 2008*). As $CB_1$ is majorly expressed in central nervous system, an average brain membrane composition of asymmetric complex membrane was selected. The membrane composition was obtained from Ingólfsson et al. and proportionally downsized according to our system (*Table 1*; *Ingólfsson et al., 2017*). 150 mM NaCl salt solution with TIP3P water model was used in the extracellular and intracellular regions (*Mark and Nilsson, 2001*). CHARMM36m force field was used to parameterize the protein, lipid, water, and ions (*Huang et al., 2017*).

For building the classical cannabinoid system, the modeled PDB structure was used. In this case, a classical cannabinoid HU-210 was docked into the orthosteric pocket using Autodock Vina (*Trott*

**Table 1.** Asymmetric average brain membrane composition used in MD simulation.

| Head Group | Lipid Group | Upper Leaflet | Lower Leaflet | Total |
|---|---|---|---|---|
| phosphatidylcholine | DPPC | 8 | 5 | 13 |
| | POPC | 14 | 7 | 21 |
| | DOPC | 4 | 2 | 6 |
| | PAPC | 7 | 4 | 11 |
| | PDoPC | 1 | 0 | 1 |
| phosphatidylethanolamine | POPE | 2 | 4 | 6 |
| | PAPE | 5 | 9 | 14 |
| | PDoPE | 8 | 15 | 23 |
| sphingolipid | SSM | 9 | 3 | 12 |
| | OSM | 1 | 0 | 1 |
| | NSM | 2 | 0 | 2 |
| phosphatidylserine | DPPS | 0 | 1 | 1 |
| | POPS | 0 | 6 | 6 |
| | PAPS | 0 | 6 | 6 |
| Glycolipid | GM1 | 2 | 0 | 2 |
| | GM3 | 2 | 0 | 2 |
| phosphatidylinositol | POPI | 0 | 5 | 5 |
| | PIPI | 0 | 2 | 2 |
| Ceramide | CER180 | 1 | 1 | 2 |
| Sterol | Cholesterol | 62 | 58 | 120 |
| Total | | 128 | 128 | 256 |

and Olson, 2010). The docked bound pose was selected based on best overall structure of HU-210 to the experimentally determined crystal structure of another bound classical cannabinoid (Ligand: AM841, PDB:6KPG *Hua et al., 2020*; *Figure 3—figure supplement 3*). The classical cannabinoid-bound system was built with identical complex membrane composition, salt concentration, and force field with NPS bound system.

Other classical cannabinoids (AMG-41, JWH-133, and O-1317) and NPS (AMB-FUBINACA, CUMYL-FUBINACA, 5F-AMP) bound systems were also set up. These ligands are docked into the orthosteric pocket. Best docking poses were selected based on optimizing the distance between the hydroxyl group of classical cannabinoid (linker oxygen for NPS) to S383$^{7.39}$ and the furthest tail atom distance

to W279[5.43]. These systems also have identical complex membrane composition, salt concentration, and force field with previously described systems.

## System minimization and equilibration

All ligand-bound systems are minimized and equilibrated before the production run. Ten thousand minimization steps were performed with the conjugate gradient method. Six sequential equilibration steps were carried out to stabilize the systems at 300 K temperature and 1 atm pressure for the production simulations. The systems were heated to 300 K in the NVT ensemble in the initial two stages. Each of these steps was performed for 250 ps. Langevin dynamics was used to control the temperature with additional damping and random force. The damping coefficient for the damping force was set as 1/picosecond. The Langevin dynamics was turned off for the hydrogen atoms. All the bonded hydrogen atoms were constrained with the SHAKE algorithm with default parameters of NAMD (*Ryckaert et al., 1977*). The integration time step for these two NVT ensemble equilibration was one femtosecond (fs). Harmonic constraints were used for fixing the temperature coupling of the protein residues with constraint scaling term set to 10 in the first NVT ensemble equilibration, followed by 5. Temperature coupling of lipid molecules was also restrained with harmonic force with a force constant equal to 5. CHARMM-GUI Selected Dihedral and Improper bonds were also restrained with an extra bonded term with a force constant of 500 in the first NVT ensemble equilibration, followed by 200. The non-bonded cutoff distance for the van der Waals interactions was set to be 12 Å with a switch distance of 10 Å, at which a switching function is turned on to truncate van der Waals interactions at the cutoff distance. Non-bonded interactions for three consecutively bonded atoms were excluded. The particle mesh Ewald method was implemented for electrostatics calculation with grid size 1 Å (*Darden et al., 1993*).

The next four equilibrations were performed in the NPT ensemble, where pressure was fixed to 1 atm with Langevin piston pressure control. The barostat oscillation period was set to 100 fs with a damping time 50 fs. These four NPT ensemble equilibrations were performed for 250, 500, 500, and 500 ps, respectively. The integration timestep for these equilibration steps was increased to two fs. The constraint for temperature coupling on the protein residues was decreased gradually with the constraint scaling term for the four NPT ensemble simulations set to 2.5, 1.0, 0.5, and 0.1, respectively. Similarly, the restraints on temperature coupling on the lipid molecules were also decreased gradually with force constant for the four NPT ensemble simulations set to 2, 1, 0.2, and 0.0, respectively. Furthermore, the restraints on the CHARMM-GUI Selected dihedral and improper bonds were decreased gradually with force constant for the four steps set to 100, 100, 50, and 0.0, respectively.

## Well-tempered metadynamics

Well-tempered metadynamics was implemented for finding unbinding pathways (*Barducci et al., 2008*; *Laio and Parrinello, 2002*). Simulations were performed with the Collective variables module (Colvars) of NAMD v2.14 (*Fiorin et al., 2013*; *Phillips et al., 2005*). In metadynamics, a history-dependent biasing potential ($V_{meta}(S, t)$) is added to the Hamiltonian of the MD simulation, which discourages the system from revisiting configurations that have already been sampled (*Barducci et al., 2011*; *Valsson et al., 2016*). The $V_{meta}(S, t)$ is a sum of Gaussians deposited along the system trajectory in the CVs space $S = (S_1(r), S_2(r) \ldots S_d(r))$ as shown in *Equation 1*, where W, $\sigma$, $\tau$ are Gaussian height, width, and deposition time step, respectively. With a sufficiently long simulation, the bias potential estimates the underlying free energy along the CVs (*Barducci et al., 2011*; *Laio and Parrinello, 2002*). The well-tempered metadynamics was introduced to increase the convergence of bias potential by decreasing the Gaussian height with time (*Equation 2*; *Barducci et al., 2011*). In the *Equation 2*, $\omega$ is the bias deposition rate and $\omega\tau$ is equivalent to constant Gaussian height for well-tempered metadynamics. The free energy is estimated from the bias potential using *Equation 3*, where the ΔT is an user-defined parameter.

$$V_{meta}(S, t) = \sum_{t'=\tau}^{t} W \prod_{i=1}^{N_{cv}} \exp(\frac{(S(r(t)) - S(r(t')))^2}{2\sigma_i^2}) \tag{1}$$

$$W = \omega\tau \exp(\frac{-V_{meta}(S, t)}{k_B \Delta T}) \tag{2}$$

$$\Delta G(\boldsymbol{S}) = -\frac{T + \Delta T}{\Delta T} V_{meta}(\boldsymbol{S}, t) \tag{3}$$

In this work, we selected two collective variables (CVs) for NPS (MDMB-FUBINACA) and classical cannabinoid (HU-210) unbinding according to *Mahinthichaichan et al., 2021*. The first collective variable is the z-direction distance from the converged toggle switch to the center of mass of all heavy carbon atoms of ligands. The second collective variable is the coordination number (CN) as defined in *Equation 4* where $d_{ij}$ represents the distance from ith atom of the ligand and the alpha carbon of jth residue in the binding pocket. Selected binding pocket residues for CN calculation are shown in the *Figure 4—figure supplement 2*. The ΔT, gaussian height ($\omega\tau$), deposition time step ($\tau$) are selected as 4200K, 0.4 kcal/mol and 100 ps, respectively. The Gaussian width for the two CVs are set to be 0.5 and 0.1, respectively.

$$CN = \sum_i \sum_j \frac{1 - (d_{ij}/4.5)^8}{1 - (d_{ij}/4.5)^{16}} \tag{4}$$

## Umbrella sampling and unbiased sampling

Umbrella sampling was performed along ligand distance from TM5 to capture the unbinding process (*Figure 4—figure supplement 1*; *Kästner, 2011*). The unbinding pathway obtained from the meta-dynamics was clustered into 300 bins by dividing the selected distances from 5 to 35 Å. The center of each bin was used as the center of each window for umbrella sampling. Five independent structures were selected from each cluster to simulate five independent umbrella runs in each umbrella window. If a cluster does not contain any structure, starting structures for that window were selected from the closest clusters. A constant harmonic biased potential of 10 kcal/mol is used for each window. OpenMM v7.8 MD engine was used to run the umbrella sampling runs (*Eastman et al., 2017*). The temperature and pressure of the systems are controlled at 300K and 1 atm by the Langevin thermostats and Monte Carlo barostats. The integration timestep was chosen to be two fs. Movements of the containing hydrogen atom were constrained using HBonds commands with SHAKE (or SETTLE for water) algorithm. The cutoff distance for non-bonded interaction other than electrostatic interaction was set to 12 Å, with a switching potential at 10 Å to make the potential to zero smoothly at the cutoff. The particle weld method was used to calculate the long-range electrostatics. Each simulation was run for 20 ns.

Identical starting structures and simulation conditions (Thermostat, barostat, cutoff, electrostatic calculation method, integration timestep, and constraints on Hydrogen bond) were selected for unbiased simulations. OpenMM v7.8 simulation software was used to run simulations. Each trajectory was run for 100 ns. All the simulations were performed on the distributive computing facility folding@ home (*Beberg et al., 2009*).

## Markov state model

Markov state model (MSM) is used to estimate the thermodynamics and kinetics from the unbiased simulation (*Prinz et al., 2011*; *Noé and Fischer, 2008*). MSM characterizes a dynamic process using the transition probability matrix and estimates its relevant thermodynamics and kinetic properties from the eigendecomposition of this matrix. This matrix is usually calculated using either maximum likelihood or Bayesian approach (*Prinz et al., 2011*; *Trendelkamp-Schroer et al., 2015*). The prevalence of MSM as a post-processing technique for MD simulations was due to its reliance on only local equilibration of MD trajectories to predict the global equilibrium properties (*Husic and Pande, 2018*; *Noé and Rosta, 2019*). Hence, MSM can combine information from distinct short trajectories, which can only attain the local equilibrium (*Bowman et al., 2014*; *Wang et al., 2018*; *Shukla et al., 2015*).

The following steps are taken for the practical implementation of the MSM from the MD data (*Dutta and Shukla, 2023*; *Dutta et al., 2022a*; *Dutta et al., 2022b*; *Bansal et al., 2023*; *Mi et al., 2023*).

1. Each frame obtained from the MD simulation was featurized using features important for capturing the conformational ensemble. In this case, the unbinding process for each ligand was featurized using distances that characterize the ligand distances to the binding pocket and binding pocket conformational change. Specifically, all heavy atom distances from each of the Cα carbon atom

of all binding pocket residues were calculated (*Figure 4—figure supplement 2*). Additionally, all possible combinations of Cα carbon atom distances between all the binding pocket residues were included to capture the binding pocket motion. Feature calculations were performed with the Python library MDTraj v1.9.8 (*McGibbon et al., 2015*). The total number of features selected for MSM building of MDMB-FUBINACA and HU-210 are 297593 and 288, respectively.

2. Dimensionality reduction was performed using time-independent component analysis (TICA) (*Pérez-Hernández et al., 2013*; *Schwantes and Pande, 2013*). We found the orthogonal projections (time-independent components) with TICA, which are linear combinations of the slowest features. In tIC space, two spatially close points are kinetically close. The lag time selected for tiC building was 5 ns.

3. Clustering was performed on the tICs using k-means clustering algorithms to discretize the space into Markovian states.

4. Lag time for the MSM was calculated by estimating the shortest time at which the timescale of the slowest processes has converged to a particular value (*Figure 4—figure supplement 3*).

5. To optimize MSM based on the cluster numbers and tIC components on which clustering is performed, we calculated the VAMP-2 score from the MSM, where VAMP stands for Variational Approach for Markov Processes (*Figure 4—figure supplement 4*; *Wu and Noé, 2020*). For a reversible MSM, this score represents the summation of the square of the k slowest eigenvalues, where k is a hyperparameter. Closer the eigenvalue is to 1, the corresponding eigenvector captures a slower process. Therefore, we optimized the MSM by maximizing the VAMP-2 score.

6. To validate the Markovian property of our optimized models, Chapman–Kolmogorov test (C-K test) was performed (*Figure 4—figure supplement 5*). C-K test states that for a Markov model, the kth power of $P(\tau)$ needs to be equal transition probability matrix determined at $k\tau$ time ($P(\tau)^k \approx P(k\tau)$). We showed that differences between the elements of transition probability matrix at higher lag times remain relatively small.

Dimensionality reduction, clustering Markov state model building, and VAMP-2 calculations are performed with the pyEMMA v2.5.6 library (*Scherer et al., 2015*). The optimized MSM for MDMB-FUBINACA unbinding simulations were built with 700 clusters, 7 tiCs, and 35 ns of lag time. For HU-210, optimized MSM were built with 800 clusters, 6 tiCs, and 35 ns of lag time.

## Transition-based reweighting analysis method

Markov State Models have been extensively used to investigate the protein-ligand binding process (*Dutta et al., 2022a*; *Buch et al., 2011*; *Lawrenz et al., 2015*; *Aldukhi et al., 2020*; *Shukla et al., 2019*; *Zhao and Shukla, 2022*; *Chen et al., 2021*; *Zhao et al., 2023*). However, these studies were mainly performed for ligands with high off-rates which could be sampled using the unbiased trajectories. For ligands with low off rates, the use of reversible transition matrix would yield incorrect estimates of unbinding kinetics. Therefore, we use the TRAM (*Wu et al., 2016*; *Galama et al., 2023*) method to accurately estimate the unbinding kinetics of new psychoactive substances. TRAM is a thermodynamics and kinetics estimator method, which, unlike MSM, can combine unbiased and biased simulation data to estimate thermodynamics and kinetics. TRAM utilizes the advantages of the local equilibrium approximation of MSM and the benefits of biased simulations to enforce local equilibrium in interstate transitions where it is difficult to attain.

As the simulations are obtained from multiple ensembles (biased and unbiased), it is paramount to classify the MD frames (or the conformations) based on which ensemble it belongs to. Each ensemble represents simulations that are performed with identical Hamiltonian energy functions. Therefore, unbiased simulations are considered as one ensemble, whereas, in umbrella sampling, each biasing window is considered a single ensemble.

Like MSM, in TRAM, the conformational space is also discretized into non-overlapping states. The interstate transitions should follow the following relationship shown in the *Equation 5*, where $f_i^k$ is the local free energy of the ith state and kth ensemble. The term $e^{-f_i^k}$ is proportional to the stationary density ($\mu(x)$) of state $S_i$ in ensemble $k$. The $\mu(x)$ of each conformation ($x$) of $S_i$ is weighted with negative exponential of bias energy ($b^k(x)$) on x in ensemble $k$ ($e^{-b^k(x)}$) (*Equation 6*).

$$e^{-f_i^k} p_{ij}^k = e^{-f_j^k} p_{ji}^k \tag{5}$$

$$e^{-f_i^k} = \int_{S_i} e^{-b^k(x)} \mu(x) \tag{6}$$

To obtain kinetics and thermodynamics information from TRAM, we have to derive interstate transitions ($p_{ij}^k$) and the stationary density of the entire ensemble ($\mu(x)$), where both the terms follow normalization constraint (*Equations 7 and 8*). Therefore, there are $m^2 K + X$ unknown variables. Therefore, to solve these unknown variables, the maximum likelihood approach has been considered, where the likelihood function is defined as *Equation 9*, which is the combination of the likelihood function of MSM and local equilibrium. This maximum likelihood problem was subjected to the constraints of *Equations 5, 7, and 8*.

$$\sum_j p_{ij}^k = 1 \tag{7}$$

$$\sum_{x \in X} \mu(x) = 1 \tag{8}$$

$$L_{TRAM} = \Pi_{k=1}^K \underbrace{(\Pi_{i,j}(p_{ij}^k)^{c_{ij}^k})}_{L_{MSM}} \underbrace{(\Pi_{i=1}^m \Pi_{x \in X_i^k} e^{f_i^k - b^k(x)} \mu(x))}_{L_{LEQ}} \tag{9}$$

Wu et al. showed that the solution of this maximum-likelihood problem can be turned into system of non-linear algebraic equations (*Equations 10–12*), where $c_{ij}^k$ is count of the interstate transitions between state $S_i$ and $S_j$ in ensemble $k$. This system of equations are solved iteratively to estimate $v_i^k$ and $f_i^k$, which provides the prediction of $p_{ji}^k$ and $\mu(x)$ (*Equations 13 and 14*).

$$\sum_j \frac{c_{ij}^k + c_{ji}^k}{exp[f_j^k - f_i^k]v_j^k + v_i^k} = 1 \tag{10}$$

$$\sum_{x \in X_i} \frac{exp(f_i^k - b^k(x))}{\sum_l R_i^l exp[f_i^l - b^l(x)]} = 1 \tag{11}$$

$$R_i^k = \sum_j \frac{(c_{ij}^k + c_{ji}^k)v_j^k}{v_j^k + exp[f_i^k - f_j^k]v_i^k} + N_i^k - \sum_j c_{ji}^k \tag{12}$$

$$p_{ij}^k = \frac{c_{ij}^k + c_{ji}^k}{exp[f_j^k - f_i^k]v_j^k + v_i^k} \tag{13}$$

$$\mu(x) = \frac{1}{\sum_k R_{i(x)}^k exp[f_{i(x)}^k - b^k(x)]} \tag{14}$$

We used the Python package pyEMMA v2.5.6 for the practical implementation of TRAM (*Scherer et al., 2015*). For calculating transition counts in ensemble $k$ ($c_{ij}^k$), the lag time of 15 ns was chosen. In the implementation, we need to preprocess each trajectory into three arrays.

1. One of the arrays represents the spatial discretization of each trajectory frame, where each frame belongs to a particular state. Therefore, each element can take values from 0 to m-1 (m is the cluster number). Before the discretization of the space, time-independent component analysis was performed on the biased and unbiased data separately. The number of tIC components for each system was selected based on the number of the tIC components of optimized MSM. Each frame from the unbiased simulation is represented by the unbiased tICs, concatenated with its feature projections on the biased tICs. Similarly, each frame from the biased simulation is represented by its feature projections on the unbiased tIC, concatenated with biased tIC projection. Therefore, NPS unbinding simulations have 14 tICs, whereas classical cannabinoid unbinding simulations have 12 tICs. The number of clusters is also obtained from the optimized MSMs.
2. Another array represents the corresponding ensemble to which each trajectory frame belongs. There are 300 windows for the umbrella sampling. Therefore, there are 301 ensembles, as the unbiased simulations represent a separate ensemble.
3. Third array represents the corresponding bias potential ($b_k(x)$) that a particular frame feels if it were to be in a particular ensemble. For umbrella sampling, the biased potential is represented as *Equation 15*, where $c_k$ is selected to be 10 kcal/mol and $y_k$ is the center of each umbrella window.

$$b_k(x) = \frac{c_k}{2kT}(x - y_k)^2 \tag{15}$$

## Transition path theory

Transition path theory (TPT) analysis is applied to calculate the transition pathway and timescale between different macrostates, representing different configurational spaces in the unbinding process (*Noé et al., 2009*; *Metzner et al., 2009*) In this work, we define macrostates as a collection of Markovian states present in the area of interest in the unbinding free energy landscape. An essential concept of transition path theory is the committer probability ($q_i^+$), which is defined as the probability of any Markovian state reaching the final metastable state before it returns to the initial state. Therefore, the Markovian states present in metastable state B has a committer probability of 1. It has been shown that committer probability follows the following system of linear equation as shown in *Equation 16*, where $P_{ik}$ is the transition probability between state $S_i$ and $S_j$ as discussed in the previous section.

$$-q_i^+ + \sum_{k \notin B} p_{ik} q_k^+ = -\sum_{k \in B} p_{ik} q_k^+ \tag{16}$$

In this work, the quantity of interest from TPT is the timescale (or rate) between the metastable state transitions as shown in *Equation 17*, where $\pi_i$ is the stationary probability of state $S_i$. TPT calculations were performed by PyEMMA v2.5.6 (*Scherer et al., 2015*).

$$k_{AB} = \frac{\sum_{k \in A} \sum_{k \notin A} \pi_i p_{ik} q_k^+}{\tau \sum_{i=1}^m \pi_i (1 - q_i^+)} \tag{17}$$

## K-L divergence analysis

Kullback–Leibler divergence (K-L divergence) analysis was performed to show the structural differences in protein conformations in different macrostates (*Dutta and Shukla, 2023*; *Fleetwood et al., 2021*). In this study, this technique was used to calculate the difference in the pairwise inverse distance distributions between macrostates. Each macrostate was represented by 1000 frames that were selected proportional to their TRAM weighted probabilities. Although K-L divergence is an asymmetric measurement, for this study, we used a symmetric version of the K-L divergence by taking the average between two macrostates. Per residue contribution of K-L divergence was calculated by taking the sum of all the pairwise distances corresponding to that residue. This analysis was performed by in-house Python code.

## Trajectory analysis

Python package GetContacts is used to perform the contact calculation (*Venkatakrishnan, 2019*). Linear interaction energy analysis was performed to calculate the interaction energy between ligand and receptor using AMBERTools CPPTraj v18.01 (*Roe and Cheatham, 2013*; *de Amorim et al., 2008*). Trajectory visualization and figure generation are performed with VMD v1.9.3 (*Humphrey et al., 1996*).

## Deep learning network for allosteric prediction

Neural relational inference (NRI) network was implemented to predict allosteric dependence between the residues in the different parts of the receptors (*Zhu et al., 2022a*; *Kipf et al., 2018*). This network is a Variational autoencoder (VAE) comprising encoding and decoding parts (*Kingma and Welling, 2013*). The encoder ($q_\phi(z|x)$) takes the input Cα coordinates of protein conformations at time t ($x_t$) and tries to learn the interactions between two residues ($z_{ij}$) in the protein as a latent space. The decoder ($p_\theta(x|z)$) network try to regenerate the protein conformation at time $t + \tau$ ($x_{t+\tau}$). Similar to other VAE, the learning process maximizes the evidence lower bound (ELBO) as shown in *Equation 18*, where $p_\theta(z)$ represents the prior distribution for $z$. Here, the prior distribution is selected as default presented in the original paper, where it is represented as a categorical distribution with $K = 4$ ($P_1 = 0.91, P_2 = 0.03, P_3 = 0.03, P_4 = 0.03$).

As shown in Equation, the ELBO consists of two terms. In the first term, further mathematical derivations can show that the first term can be represented as the reconstruction error (*Equation 19*), where $\sigma^2$ is variance of the distribution, a user-defined parameter.

$$L(\phi, \theta) = \mathbb{E}_{q_\phi(z|x)}[\log p_\theta(x|z)] - KL[q_\phi(x|z)\|p_\theta(z)] \tag{18}$$

$$\mathbb{E}_{q_\phi(z|x)}[\log p_\theta(x|z)] = \sum_j \sum_{t=2}^T \frac{\| x_j^t - \mu_j^t \|^2}{2\sigma^2} + const \tag{19}$$

The second term is also called regularization term which is the K-L divergence between estimated posterior ($q_\phi(z|x)$) and prior distribution ($p_\theta(z)$) (**Equation 20**). As the prior distribution is a categorical distribution, the K-L divergence becomes entropy of the posterior distribution. We obtained the code for the NRI network from the GitHub implementation and kept most of hyperparameters as default for our training, except for decreasing the hidden layer size to 64 (**Zhu et al., 2022b**). From each unbinding simulations, 10 unbiased trajectories were selected where the ligand remain in the bound pose. Each trajectory has a length of 100 ns. In both cases, the $\tau$ was selected to be 5 ns. The allosteric weights (posterior probability) were obtained from the validation data (2 trajectories), where training was performed with remaining eight trajectories (**Figure 8—figure supplement 2**). This procedure was repeated three times, where training and validation data were selected randomly.

$$KL[q_\phi(z|x)\|p_\theta(z)] = \sum_{i \neq j} H(q_\phi(z_{ij}|x)) + const \tag{20}$$

## Mutual information estimation

Mutual information between dynamics of residue pairs was computed based on the backbone dihedral angles, as this provides a metric that is independent of the relative distances between residues. The calculations were done on same trajectory data as NRI analysis. Python package MDEntropy was used for estimating mutual information between backbone dihedral angles of two residues (**Hernández and Pande, 2017**).

## Standard binding free energy calculations

To calculate the standard binding free energy from simulation, we adopted a procedure described in **Buch et al., 2011**. In this procedure, a volumetric correction term is added to the PMF to calculate the final binding free energy (**Equation 21**). The volumetric correction term is used to predict the free energy at the standard condition (1M) as shown in **Equation 22**, where $V_o$ corresponds to the volume of a molecule should occupy at the standard condition and $V_u$ is the volume of the unbounded state in the simulation box. The expression for the PMF contribution of the free energy is shown is in **Equation 23**, where the denominator of the equation can be represented as $\int_u exp(-\beta W(r))dr = exp(-\beta \Delta W)V_u$.

Therefore, the final derivation of $\Delta G$ is shown in **Equation 24**.

$$\Delta G = \Delta G_{pmf} + \Delta G_V \tag{21}$$

$$\Delta G_V = -k_B T \log(\frac{V_u}{V_o}) \tag{22}$$

$$\Delta G_{pmf} = -k_B T \log \frac{\int_b exp(-\beta W(r))dr}{\int_u exp(-\beta W(r))dr} \tag{23}$$

$$\Delta G = -k_B T \log \frac{\int_b exp(-\beta W(r))dr}{V_o} - \Delta W \tag{24}$$

In this work, to estimate the free energy ($\Delta G$) x, y, z component of the ligand center of mass is calculated compared to the center of mass of the alpha carbons of binding pocket residues. The three-dimensional space was discretized into $25 \times 25 \times 50$ bins and each bin is weighted using TRAM-calculated probability density. Depth of the pmf ($\Delta W$) was calculated by averaging the pmf of the 100 bins with highest pmf in the bulk. To evaluate the weighted binding volume ($\int_b exp(-\beta W(r))dr$), we selected the bins with pmf less than 1 kcal/mol.

## Acknowledgements

DS acknowledges support from NIGMS MIRA award R35GM-142745 and NSF Early CAREER Award (MCB-1845606). SD and DS thank folding@home donors for providing computational resources for the study.

## Additional information

### Funding

| Funder | Grant reference number | Author |
|---|---|---|
| National Institute of General Medical Sciences | NIGMS MIRA award R35GM-142745 | Diwakar Shukla |
| National Science Foundation | NSF Early CAREER Award MCB-1845606 | Diwakar Shukla |

The funders had no role in study design, data collection and interpretation, or the decision to submit the work for publication.

### Author contributions

Soumajit Dutta, Conceptualization, Formal analysis, Investigation, Methodology, Writing - original draft, Writing – review and editing; Diwakar Shukla, Conceptualization, Resources, Supervision, Funding acquisition, Project administration, Writing – review and editing

### Author ORCIDs

Soumajit Dutta  http://orcid.org/0000-0003-3951-8088
Diwakar Shukla  https://orcid.org/0000-0003-4079-5381

Reviewer #1 (Public review): https://doi.org/10.7554/eLife.98798.3.sa1
Reviewer #2 (Public review): https://doi.org/10.7554/eLife.98798.3.sa2
Author response https://doi.org/10.7554/eLife.98798.3.sa3

## Additional files

### Supplementary files

MDAR checklist

### Data availability

Unbinding simulation trajectories and topology files that have been used for the analysis has been deposited in Dryad and can be obtained from https://doi.org/10.5061/dryad.4f4qrfjq5. Python scripts and necessary files to generate the figures is provided in the github repository (https://github.com/ShuklaGroup/Dutta_Shukla_Cannabinoid_2023a copy archived at *Dutta, 2025*).

The following dataset was generated:

| Author(s) | Year | Dataset title | Dataset URL | Database and Identifier |
|---|---|---|---|---|
| Dutta S, Shukla D | 2025 | Data from: Characterization of binding kinetics and intracellular signaling of new psychoactive substances targeting cannabinoid receptor using transition-based reweighting method | https://doi.org/10.5061/dryad.4f4qrfjq5 | Dryad Digital Repository, 10.5061/dryad.4f4qrfjq5 |

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
