## [Editor Report · eLife Assessment]

A combination of molecular dynamics simulation and state-of-the-art statistical post-processing techniques provided **valuable** insight into GPCR-ligand dynamics. This manuscript provides **solid** evidence for differences in the binding/unbinding of classical cannabinoid drugs from new psychoactive substances. The results could aid in mitigating the public health threat these drugs pose.

---

## [Referee Report · Reviewer #1 (Public review)]

This manuscript presents insights into biased signaling in GPCRs, namely cannabinoid receptors. Biased signaling is of broad interest in general, and cannabinoid signaling is particular relevant for understanding the impact of new drugs that target this receptor. Mechanistic insight from work like this could enable new approaches to mitigate the public health impact of new psychoactive drugs. Towards that end, this manuscript seeks to understand how new psychoactive substances (NPS, e.g. MDMB-FUBINACA) elicit more signaling through β-arrestin than classical cannabinoids (e.g. HU-210). The authors use an interesting combination of simulations and machine learning.

The caption for Figure 3 doesn't explain the color scheme, so its not obvious what the start and end states of the ligand are.

For the metadynamics simulations were multiple Gaussian heights/widths tried to see what, if any, impact that has on the unbinding pathway? That would be useful to help ensure all the relevant pathways were explored.

It would be nice to acknowledge previous applications of metadynamics+MSMs and (separately) TRAM, such as Simulation of spontaneous G protein activation... (Sun et al. eLife 2018) and Estimation of binding rates and affinities... (Ge and Voelz JCP 2022).

What is KL divergence analysis between macrostates? I know KL divergence compares probability distributions, but its not clear what distributions are being compared.

I suggest being more careful with the language of universality. It can be "supported" but "showing" or "proving" its universal would require looking at all possible chemicals in the class.

Comments on revisions:

The authors provided appropriate responses to the comments above.

---

## [Referee Report · Reviewer #2 (Public review)]

Summary:

The investigation provides a computational as well as biochemical insights into the (un)binding mechanisms of a pair of psychoactive substances into cannabinoid receptors. A combination of molecular dynamics simulation and a set of state-of-the art statistical post-processing techniques were employed to exploit GPCR-ligand dynamics.

Strengths:

The strength of the manuscript lies in usage and comparison of TRAM as well as Markov state modelling (MSM) for investigating ligand binding kinetics and thermodynamics. Usually MSMs have been more commonly used for this purpose. But as the authors have pointed out, implicit in the usage of MSMs lie the assumption of detailed balance, which would not hold true for many cases especially those with skewed binding affinities. In this regard, the author's usage of TRAM which harnesses both biased and unbiased simulations for extracting the same, provides a more appropriate way-out.

Weaknesses:

(1) While the authors have used TRAM (by citing MSM to be inadequate in these cases), the thermodynamic comparisons of both techniques provide similar values. In this case, one would wonder what advantage TRAM would hold in this particular case.

(2) The initiation of unbiased simulations from previously run biased metadynamics simulations would almost surely introduce hysteresis in the analysis. The authors need to address these issues.

(3) The choice of ligands in the current work seems very forced and none of the results compare directly with any experimental data. An ideal case would have been to use the seminal D.E. Shaw research paper on GPCR/ligand binding as a benchmark and then show how TRAM, using much lesser biased simulation times, would fare against the experimental kinetics or even unbiased simulated kinetics of the previous report

(4) The method section of the manuscript seems to suggest all the simulations were started from a docked structure. This casts doubt on the reliability of the kinetics derived from these simulations that were spawned from docked structure, instead of any crystallographic pose. Ideally, the authors should have been more careful in choosing the ligands in this work based on the availability of the crystallographic structures.

(5) The last part of using a machine learning-based approach to analyse allosteric interaction seems to be very much forced, as there are numerous distance-based more traditional precedent analyses that do a fair job of identifying an allosteric job.

(6) While getting busy with the methodological details of TRAM vs MSM, the manuscript fails to share with sufficient clairty what the distinctive features of two ligand binding mechanisms are.

Comments on revisions:

The authors have addressed most of the queries of the reviewer in an adequate manner. However, The current code availability section just provides the link to Python files to generate the plots. It is not very useful in its current form. The code availability section should provide a proper GitHub page that shows the usage of TRAM for the readers to execute. While Pyemma has been cited for TRAM, a python note book to reproduce the TRAM would be very instructive.

---

## [Author Response]

The following is the authors’ response to the original reviews

**Public Reviews:**

**Reviewer #1 (Public Review):**
This manuscript presents insights into biased signaling in GPCRs, namely cannabinoid receptors. Biased signaling is of broad interest in general, and cannabinoid signaling is particularly relevant for understanding the impact of new drugs that target this receptor. Mechanistic insight from work like this could enable new approaches to mitigate the public health impact of new psychoactive drugs. Towards that end, this manuscript seeks to understand how new psychoactive substances (NPS, e.g. MDMB-FUBINACA) elicit more signaling through βarrestin than classical cannabinoids (e.g. HU-210). The authors use an interesting combination of simulations and machine learning.

We thank the reviewer for the comments. We have provided point by point response to the reviewer’s comment below and incorporated the suggestions in our revised manuscript. Modified parts of manuscripts are highlighted in yellow.

Comments:(1) The caption for Figure 3 doesn't explain the color scheme, so it's not obvious what the start and end states of the ligand are.

We thank the reviewer to point this out. We have added the color scheme in the figure caption.

(2) For the metadynamics simulations were multiple Gaussian heights/widths tried to see what, if any, impact that has on the unbinding pathway? That would be useful to help ensure all the relevant pathways were explored.

We thank the reviewer for the suggestion. We agree with the reviewer that gaussian height/width may impact unbinding pathway. However, we like to point out that we used a well-tempered version of the metadynamics. In well-tempered metadynamics, the effective gaussian height decreases as bias deposition progresses. Therefore, we believe that the gaussian height/width should have minimal impact on the unbinding pathway. To address the reviewer's suggestion, we conducted additional well-tempered metadynamics simulations varying key parameters such as bias height, bias factor, and the deposition rate, all of which can influence the sampling space. Parameter values for bias height, bias factor and deposition rate that we originally used in the paper are 0.4 kcal/mol, 15 and 1/5 ps^-1^, respectively. We explored different values for these parameters and projected the sampled space on top of previously sampled region (Figure S4). We observed that new simulations sample similar unbinding pathway in the extracellular direction and discover similar space in the binding pocket as well.

Results and Discussion (Page 10)

“We also performed unbinding simulations using well-tempered metadynamics parameters (bias height, bias deposition rate and bias factor) to confirm the existence of alternative pathways (Figure S4). However, the simulations show that ligands follow the similar pathway for all

metadynamics runs.”

(3) It would be nice to acknowledge previous applications of metadynamics+MSMs and (separately) TRAM, such as the Simulation of spontaneous G protein activation... (Sun et al. eLife 2018) and Estimation of binding rates and affinities... (Ge and Voelz JCP 2022).

We appreciate the reviewer's feedback. We have incorporated additional citations of studies demonstrating the use of TRAM as an estimator for both kinetics and thermodynamics (e.g. Ligand binding: Ge, Y. and Voelz, V.A., JCP, 2022[1]; Peptide-protein binding kinetics: Paul, F. et al., Nat. Commun., 2017[2], Ge, Y. et al., JCIM, 2021[3]). Additionally, we have included references to studies where biased simulations were initially used to explore the conformational space, and the results were then employed to seed unbiased simulations for building a Markov state model. (Metadynamics: Sun, X. et al., elife, 2018[4]; Umbrella Sampling: Abella, J. R. et al., PNAS, 2020[5]; Replica Exchange: Paul, F. et al., Nat. Commun., 2017[2]).

(4) What is KL divergence analysis between macrostates? I know KL divergence compares probability distributions, but it is not clear what distributions are being compared.

We apologize for this confusion. The KL divergence analysis was performed on the probability distributions of the inverse distances between residue pairs from any two macrostates. Each macrostate was represented by 1000 frames that were selected proportional to the TRAM stationary density. All possible pair-wise inverse distances were calculated per frame for the purpose of these calculations. Although KL divergence is inherently asymmetric, we symmetrized the measurement by calculating the average. Per-residue K-L divergence, which is shown in the main figures as color and thickness gradient, was calculated by taking the sum of all pairs corresponding to the residue. We have included a detailed discussion of K-L divergence in Methods section. We have also modified the result section to add a brief discussion of K-L divergence methodology.

Results and Discussion (Page 15)

“We further performed Kullback-Leibler divergence (K-L divergence) analysis between inverse distance of residue pairs of two macrostates to highlight the protein region that undergoes high conformational change with ligand movement.”

Methods (Page 33)

“Kullback–Leibler divergence (K-L divergence) analysis was performed to show the structural differences in protein conformations in different macrostates[4,114] . In this study, this technique was used to calculate the difference in the pairwise inverse distance distributions between macrostates. Each macrostate was represented by 1000 frames that were selected proportional to their TRAM weighted probabilities. Although K-L divergence is an asymmetric measurement, for this study, we used a symmetric version of the K-L divergence by taking the average between two macrostates. Per residue contribution of K-L divergence was calculated by taking the sum of all the pairwise distances corresponding to that residue. This analysis was performed by inhouse Python code.”

(5) I suggest being more careful with the language of universality. It can be "supported" but "showing" or "proving" its universal would require looking at all possible chemicals in the class.

We thank the reviewer for the suggestion. In response, we have revised the manuscript to ensure that the language reflects that our findings are based on observations from a limited set of ligands, namely one NPS and one classical cannabinoid. We have replaced references to ligand groups (such as NPS or classical cannabinoid) with the specific ligand names (such as MDMB-FUBINACA or HU-210) to avoid claims of universality and prevent any potential confusion.

Results and Discussion (Page 19)

“In this work, we trained the network with the NPS (MDMB-FUBINACA), and classical cannabinoid (HU-210) bound unbiased trajectories (Method Section). Here, we compared the allosteric interaction weights between the binding pocket and the NPxxY motif which involves in triad interaction formation. Results show that each binding pocket residue in MDMBFUBINACA bound ensemble shows higher allosteric weights with the NPxxY motif, indicating larger dynamic interactions between the NPxxY motif and binding pocket residues(Figure S9). The probability of triad formation was estimated to observe the effect of the difference in allosteric control. TRAM weighted probability calculation showed that MDMB-FUBINACA bound CB1 has the higher probability of triad formation (Figure 8A). Comparison of the pairwise interaction of the triad residues shows that interaction between Y397^7.53^-T210^3.46^ is relatively more stable in case of MDMB-FUBINACA bound CB1, while other two inter- actions have similar behavior for both systems (Figures S10A, S10B, and S10C). Therefore, higher interaction between Y397^7.53^ and T210^3.46^ in MDMB-FUBINACA bound receptor causes the triad interaction to be more probable.

Furthermore, we also compared TM6 movement for both ligand bound ensemble which is another activation metric involved in both G-protein and β-arrestin binding. Comparison of TM6 distance from the DRY motif of TM3 shows similar distribution for HU-210 and MDMBFUBINACA (Figure 8B). These observations support that NPS binding causes higher β-arrestin signaling by allosterically controlling triad interaction formation.”

**Reviewer #2 (Public Review):**
Summary:The investigation provides computational as well as biochemical insights into the (un)binding mechanisms of a pair of psychoactive substances into cannabinoid receptors. A combination of molecular dynamics simulation and a set of state-of-the art statistical post-processing techniques were employed to exploit GPCR-ligand dynamics.Strengths:The strength of the manuscript lies in the usage and comparison of TRAM as well as Markov state modelling (MSM) for investigating ligand binding kinetics and thermodynamics. Usually, MSMs have been more commonly used for this purpose. But as the authors have pointed out, implicit in the usage of MSMs lies the assumption of detailed balance, which would not hold true for many cases especially those with skewed binding affinities. In this regard, the author's usage of TRAM which harnesses both biased and unbiased simulations for extracting the same, provides a more appropriate way out.Weaknesses:(1) While the authors have used TRAM (by citing MSM to be inadequate in these cases), the thermodynamic comparisons of both techniques provide similar values. In this case, one would wonder what advantage TRAM would hold in this particular case.

We thank the reviewer for the comment. While we agree that the thermodynamic comparisons between MSM and TRAM provide similar values in this instance, we would like to emphasize the underlying reasoning behind our choice of TRAM.

MSM can struggle to accurately estimate thermodynamic and kinetic properties in cases where local state reversibility (detailed balance) is not easily achieved with unbiased sampling. This is especially relevant in ligand unbinding processes, which often involve overcoming high free energy barriers. TRAM, by incorporating biased simulation data (such as umbrella sampling) in addition to unbiased data, can better achieve local reversibility and provide more robust estimates when unbiased sampling is insufficient.

The similarity in thermodynamic estimates between MSM and TRAM in our study can be attributed to the relatively long unbiased sampling period (> 100 µs) employed. With sufficient sampling, MSM can approach detailed balance, leading to results comparable to those from TRAM. However, as we demonstrated in our manuscript (Figure 4D), when the amount of unbiased sampling is reduced, the uncertainties in both the thermodynamics and kinetics estimates increase significantly for MSM compared to TRAM. Thus, while MSM and TRAM perform similarly under the conditions of extensive sampling, TRAM's advantage lies in its robustness when unbiased sampling is limited or difficult to achieve.

(2) The initiation of unbiased simulations from previously run biased metadynamics simulations would almost surely introduce hysteresis in the analysis. The authors need to address these issues.

We thank the reviewer for the comment. We acknowledge that biased simulations could potentially introduce hysteresis or result in the identification of unphysical pathways. However, we believe this issue is mitigated using well-tempered metadynamics, which gradually deposit a decaying bias. This approach enables the simulation to explore orthogonal directions of collective variable (CV) space, reducing the likelihood of hysteresis effects(Invernizzi, M. and Parrinello, M., JCTC, 2019[6]).

Furthermore, there is precedent for using metadynamics-derived pathways to initiate unbiased simulations for constructing Markov State Models (MSMs). This methodology has been successfully applied in studying G-protein activation (Sun, X. et al., elife, 2018[4]).

Additional support to our observation can be found in two independent binding/unbinding studies of ligands from cannabinoid receptors, which have discovered similar pathway using different CVs (Saleh, et al., Angew. Chem., 2018[7]; Hua, T. et al., Cell, 2020[8]).

(3) The choice of ligands in the current work seems very forced and none of the results compare directly with any experimental data. An ideal case would have been to use the seminal D.E. Shaw research paper on GPCR/ligand binding as a benchmark and then show how TRAM, using much lesser biased simulation times, would fare against the experimental kinetics or even unbiased simulated kinetics of the previous report

We would like to address the reviewer's concerns regarding the choice of ligands, lack of direct experimental comparison, and the use of TRAM, and clarify our rationale point by point:

Ligand Choice: The ligands selected for this study were chosen due to their relevance and well characterized binding properties. MDMB-FUBINACA is well-known NPS ligand with documented binding properties. This ligand is still the only NPS ligand with experimentally determined CB1 bound structure (Krishna Kumar, K. et al., Cell, 2019[9]). Similarly, the classical cannabinoid (HU-210) used in this study has established binding characteristics and is one of earliest known synthetic classical cannabinoid. Therefore, these ligands serve as representative compounds within their respective categories, making them suitable for our comparative analysis.

Experimental Comparison: We have indeed compared our simulation results to experimental data, particularly focusing on binding free energies. In the result section, we have shown that the relative binding free energy estimated from our simulation aligns closely with the experimentally measured values. Additionally, Absolute binding energy estimates are also within ~3 kcal/mol of the experimentally predicted value.

TRAM Performance: TRAM estimated free energies, and rates have been benchmarked against experimental predictions for various studies along with our study (Peptide-protein binding: Paul, F. et al., Nat. Commun., 2017[2]; Ligand unbinding: Wu, H. et al., PNAS, 2016[10]) . As the primary goal of this study is to compare ligand unbinding mechanism, we believe benchmarking against other datasets, such as the D.E. Shaw GPCR/ligand binding paper, is not essential for this work.

(4) The method section of the manuscript seems to suggest all the simulations were started from a docked structure. This casts doubt on the reliability of the kinetics derived from these simulations that were spawned from docked structure, instead of any crystallographic pose. Ideally, the authors should have been more careful in choosing the ligands in this work based on the availability of the crystallographic structures.

We thank the reviewer for the comment. We would like to clarify that we indeed used an experimentally derived pose for one of the ligands (MDMB-FUBINACA) as the cryo-EM structure of MDMB-FUBINACA bound to the protein was available (PDB ID: 6N4B) (Krishna Kumar K. et al., Cell, 2019[9]). However, as the cryo-EM structure had missing loops, we modeled these regions using Rosetta. We apologize for this confusion and have modified our method section to make this point clearer.

Regarding HU-210, we acknowledge that a crystallographic or cryo-EM structure for this specific ligand was not available. We selected HU-210 because it is most commonly used example of classical cannabinoid in the literature with extensively studied thermodynamic properties. Importantly, our docking results for HU-210 align closely with previously experimentally determined poses for other classical cannabinoids (Figure S11) and replicate key polar interactions, such as those with S383^7.39^, which are characteristic of this class of compounds.

System Preparation (Page 22)

“Modeling of this membrane proximal region was also performed Remodel protocol of Rosetta loop modeling. A distance constraint is added during this modeling step between C98N−term and C107N−term to create the disulfide bond between the residues. [74,76]

As the cryo-EM structure of MDMB-FUBINACA was known, ligand coordinate of MDMB- FUBINACA was added to the modeled PDB structure. The “Ligand Reader & Modeler” module of CHARMM-GUI was used for ligand (e.g., MDMB-Fubinaca) parameterization using CHARMM General Force Field (CGenFF).[77]”

(5) The last part of using a machine learning-based approach to analyze allosteric interaction seems to be very much forced, as there are numerous distance-based more traditional precedent analyses that do a fair job of identifying an allosteric job.

We thank the reviewer for the valuable comment. Neural relational inference method, which leverages a VAE (Variational Autoencoder) architecture, attempts to reconstruct the conformation (X) at time t + τ based on the conformation at time t. In doing so, it captures the non-linear dynamic correlations between residues in the VAE latent space. We chose this method because it is not reliant on specific metrics such as distance or angle, making it potentially more robust in predicting allosteric effects between the binding pocket residues and the NPxxY motif.

In response to the reviewer's suggestion, we have also performed a more traditional allosteric analysis by calculating the mutual information between the binding pocket residues and the NPxxY motif. Mutual information was computed based on the backbone dihedral angles, as this provides a metric that is independent of the relative distances between residues. Our results indicate that the mutual information between the binding pocket residues and the NPxxY motif is indeed higher for the NPS binding simulation (Figure S11).

Method

Mutual information calculation

Mutual information was calculated on same trajectory data as NRI analysis. Python package MDEntropy was used for estimating mutual information between backbone dihedral angles of two residues.

Results and Discussion (Page 21)

“To further validate our observations, we estimated allosteric weights between the binding pocket and the NPxxY motif by calculating mutual information between residue movements. Mutual information analysis reaffirms that allosteric weights between these residues are indeed higher for the MDMB-FUBINACA bound ensemble (Figure S11).”

Mutual Information Estimation (Page 37)

“Mutual information between dynamics of residue pairs was computed based on the backbone dihedral angles, as this provides a metric that is independent of the relative distances between residues. The calculations were done on same trajectory data as NRI analysis. Python package MDEntropy was used for estimating mutual information between backbone dihedral angles of two residues.[124]”

(6) While getting busy with the methodological details of TRAM vs MSM, the manuscript fails to share with sufficient clarity what the distinctive features of two ligand binding mechanisms are.

We thank the reviewer for the insightful comment. In the manuscript, we discussed that the overall ligand (un)binding pathways are indeed similar for both ligands. Therefore, they interact with similar residues during the unbinding process. However, we have focused on two key differences in unbinding mechanism between the two ligands:

(1) MDMB-FUBINACA exhibits two distinct unbinding mechanisms. In one, the linked portion of the ligand exits the receptor first. In the other mechanism, the ligand rotates within the pocket, allowing the tail portion to exit first. By contrast, for HU-210, we observe only a single unbinding mechanism, where the benzopyran ring leads the ligand out of the receptor. We have highlighted these differences in the Figure 6 and 7 and talked about the intermediate states appear along these different unbinding mechanisms. For further clarification of these differences, we have added arrows in the free energy landscapes to highlight these distinct pathways.

(2) In the bound state, a significant difference is observed in the interaction profiles. HU-210, a classical cannabinoid, forms strong polar interactions with TM7, while MDMB-FUBINACA shows weaker polar interactions with this region.

We have discussed these differences in the Results and Discussion section (Page 13-18) & conclusion section (Page 23-24).

**Recommendations for the authors:**

**Reviewer #2 (Recommendations For The Authors):**
(1) The authors should choose at least one case where the ligand's crystallographic pose is known and show how TRAM works in comparison to MSM or experimental report.

We thank the reviewer for the comment. We have used the experimentally determined cryo-EM pose for one of the ligands (i.e. MDMB-FUBINACA). We have modified the manuscript to avoid confusion. (Please refer to the response of comment 4 of reviewer 2)

(2) The authors should consider existing traditional methods that are used to detect allostery and compare their machine-learning-based approach to show its relevance.

We appreciate the reviewer’s comment. We have performed the traditional analysis by calculating mutual information between residue dynamics. We have shown that the traditional analysis matches with Machine learning based NRI calculation. (Please refer to the response of comment 5 of reviewer 2)

(3) Figure 3 doesn't provide a guide on the pathway of ligand. Without a proper arrow, it is difficult to surmise what is the start and end of the pathway. The figures should be improved.

We appreciate the reviewer’s suggestion. In response, we have revised Figure 3 to clearly indicate the ligand’s unbinding pathway by adding directional arrows and labeling the bound pose. Additionally, we have updated the figure caption to better clarify the color scheme used in the illustration.

(4) The Figure 5 presentation of free energetics has a very similar shape for the two ligands. More clarity is required on how these two ligands are different.

We thank the reviewer for the comment. While the overall shapes of the free energy profiles for the two ligands are indeed similar, this is expected as both ligands dissociate from the same pocket and follow a comparable pathway. However, key differences in their unbinding mechanisms arise due to variations in the ligand motion within the pocket. Specifically, the intermediate metastable minima in the free energy landscapes reflect these differences. For instance, in the NPS unbinding free energy landscape, the intermediate metastable state I1 corresponds to a conformation where the NPS ligand maintains a polar interaction with TM7, while the tail of the ligand has shifted away from TM5. This intermediate state is absent in the classical cannabinoid unbinding pathway, where no equivalent conformation appears in the landscape.

(6) Page 30: TICA is wrongly expressed as 'Time-independent component analysis'. It is not a time-independent process. Rather it is 'Time structured independent component analysis'.

We thank the reviewer for pointing this out. TICA should be expressed as Time-lagged independent component analysis or Time-structure independent component analysis. We have used the first expression and modified the manuscript accordingly.

(7) The manuscript's MSM theory part is quite well-known which can be removed and appropriate papers can be cited.

We thank the reviewer for the comment. We have removed the theory discussion of MSM and cited relevant papers.

“Markov State Model

Markov state model (MSM) is used to estimate the thermodynamics and kinetics from the unbiased simulation.[56,91] MSM characterizes a dynamic process using the transition probability matrix and estimates its relevant thermodynamics and kinetic properties from the eigendecomposition of this matrix. This matrix is usually calculated using either maximum likelihood or Bayesian approach.[56,97] The prevalence of MSM as a post-processing technique for MD simulations was due to its reliance on only local equilibration of MD trajectories to predict the global equilibrium properties.[92,93] Hence, MSM can combine information from distinct short trajectories, which can only attain the local equilibrium.[94–96]

The following steps are taken for the practical implementation of the MSM from the MD data. [4,17,98–100]”

(8) A proper VAMP score-based analysis should be provided to show confidence in MSM's clustering metric and other hyperparameters.

We thank the reviewer for the recommendation. VAMP-2 score based analysis had been discussed in the method section. We estimated VAMP-2 score of MSM built with different cluster number and input TIC dimensions (Figure S15). Model with best VAMP-2 was selected for comparison with TRAM result.